# Gestational choline supplementation regulates hippocampal granule neuron development and emotion-like behavior

Xiaohui Shi [1,2,3], Yuanyuan Li[2], Mengjiao Wang[1,2], Xinlan Zong[1,2], Min Liang[1,2], Quan Li[4], Weiwei Xiao[1,2], Zhongsheng Sun [1,5] ✉ & Yan Wang [1,2] ✉

Prenatal nutrition can profoundly influence the brain development of offspring through mechanisms including epigenetic reprogramming. However, the complex interplay of maternal nutrition, neural development, and behaviors at the cellular level remains to be explored. Here, we report that gestational choline supplementation (GCS) in mice attenuates anxiety- and depression-like behaviors in male offspring (F1$_{GCS}$). Multi-omics analysis reveals the comprehensive roles of GCS in diverse aspects of neuronal functions in F1$_{GCS}$. Notably, the transcriptome, intercellular communication, and chromatin accessibility of immature granule neurons were altered in the hippocampus of F1$_{GCS}$. This was accompanied by enhanced transcription and increased chromatin accessibility in glutamate signaling and suppressed transcription and reduced chromatin accessibility in MIF pathways in these neurons. Correlation analysis of cell populations with neuropsychiatric disorders showed a strong association between immature granule neurons and emotional disorders. Together, these findings provide molecular and cellular evidence supporting the effects of a widely used nutritional supplement in nutritional interventions.

Suboptimal gestational nutrition has been reported to affect the development and health of offspring in mammals[1]. In humans, for example, poor maternal nutrition is associated with low birth weight and an increased risk of metabolic diseases over several generations[2]. Under controlled experimental setups, parental protein-restricted diets[3], chronic high-fat diets[4], and maternal undernourishment reprogram[5] affect metabolic systems in rodent offspring. Prenatal nutrition particularly exerts a crucial role in shaping the developing brain and nervous system[5]. Therefore, evidence-based intervention in maternal nutrition during gestation is well-recognized and widely implemented to benefit the physical and mental health of offspring. However, our understanding of the effect of maternal nutrition on the neural development of offspring at the cellular level remains limited, thus requiring advanced techniques to unravel the complex interplay of early-life exposures and development and to bridge the gap between population-level observations and cellular-level mechanisms.

Choline, a widely used nutritional supplement, plays essential roles in cell-membrane structural integrity, methyl metabolism, and cholinergic signaling[6]. Given its critical roles in physiological systems and early brain development, the effects of maternal choline supplementation on offspring

have been extensively explored[7,8]. In humans, for example, high maternal dietary choline intake, mimicking the effects of gestational folic acid supplement, is associated with a reduced risk of neural tube defects in infants[9,10]. In rodents, gestational choline supplementation (GCS) permanently enhances long-term potentiation, attention, memory, and cognition in the offspring[11,12]. Although numerous studies support the beneficial effects of GCS on neural plasticity and memory, the intergenerational influence of GCS on emotion-like behaviors remains poorly characterized. Nevertheless, large-scale, population-based studies have suggested a negative correlation between choline concentration and the prevalence of anxiety/depression symptoms[13–15]. However, whether GCS influences anxiety- and depression-like behaviors and the underlying molecular mechanisms remain poorly characterized.

Granule neurons are a type of neuron distributed across multiple brain regions, including the hippocampus, cerebellar cortex, and olfactory bulb. Within the hippocampus, the granule neurons constitute the predominant neuronal population in the dentate gyrus and experience a pronounced developmental peak during the perinatal period[16,17]. Due to its pivotal role in mediating essential neural functions, dysregulation of granule cell activity

[1]Interdisciplinary Science Center, State Key Laboratory of Animal Biodiversity Conservation and Integrated Pest Management, Institute of Zoology, Chinese Academy of Sciences, Beijing, China. [2]College of Life Sciences, University of the Chinese Academy of Sciences, Beijing, China. [3]Medical AI Lab, The First Hospital of Hebei Medical University, Hebei Medical University, Shijiazhuang, China. [4]School of Life Sciences, Hebei University, Baoding, China. [5]Beijing Children's Hospital, National Center for Children's Health, Beijing, China. ✉e-mail: sunzs@biols.ac.cn; yanwang@ioz.ac.cn

has been linked to several neurological and psychiatric disorders, including Alzheimer's disease, epilepsy, and major depressive disorder in both mouse models[18,19] and humans[20–22]. The predominant population of granule neurons is established during embryonic development (E14.5–18.5 in mice), when immature granule neurons undergo radial migration to populate the developing dentate gyrus. Environmental stimuli, such as physical exercise, enriched environments and nutritional factors, have been shown to enhance adult neurogenesis in the dentate gyrus[23]. This process has been shown to inhibit the activity of mature neurons, thereby attenuating anxiety-like behaviors[24]. However, whether GCS regulates granule cell maturation and the subsequent influence on complex behavior remain to be addressed.

Single-cell omics technology has facilitated the uncovering of dynamic characteristics of the hippocampal neurogenic niche at both cellular and molecular levels[25–31]. These technologies also enable in-depth exploration of the complex molecular interactions underlying various cell types within the hippocampus[32]. In this study, we systematically investigated the intergenerational effects of GCS on the emotion-like behaviors of offspring by performing a series of depression- and anxiety-like behavioral tests. To explore the underlying mechanisms, GCS-induced alterations of gene transcription and chromatin modification in the hippocampus of offspring

on a genome-wide scale were profiled via a comprehensive analysis combining single-nucleus transcriptome, single-nucleus epigenome, and bulk transcriptome. Notably, GCS was observed to have a significant impact on the transcriptome, intercellular communication, and chromatin accessibility of the immature granule neurons in the hippocampus, which were associated with the reduction in the anxiety- and depression-like behaviors of GCS offspring. Our study provides comprehensive evidence demonstrating how a widely used nutritional supplement supports optimal brain function and behavior. The finding not only expands the potential therapeutic applications of choline but also enhance our understanding of the molecular and cellular underpinnings of nutritional interventions.

## Results

### GCS promotes anxiolytic and anti-depressive-like responses to acute stress in offspring

To examine the effects of GCS on emotion-like behaviors, pregnant female mice were supplemented with choline (5 g/kg choline chloride; GCS) or fed a basal diet (1.1 g/kg choline chloride; CON) from gestational day 9 till delivery in accordance with established protocols on choline supplementation across various mouse models[12,33] (Fig. 1A). At 2 months old, the

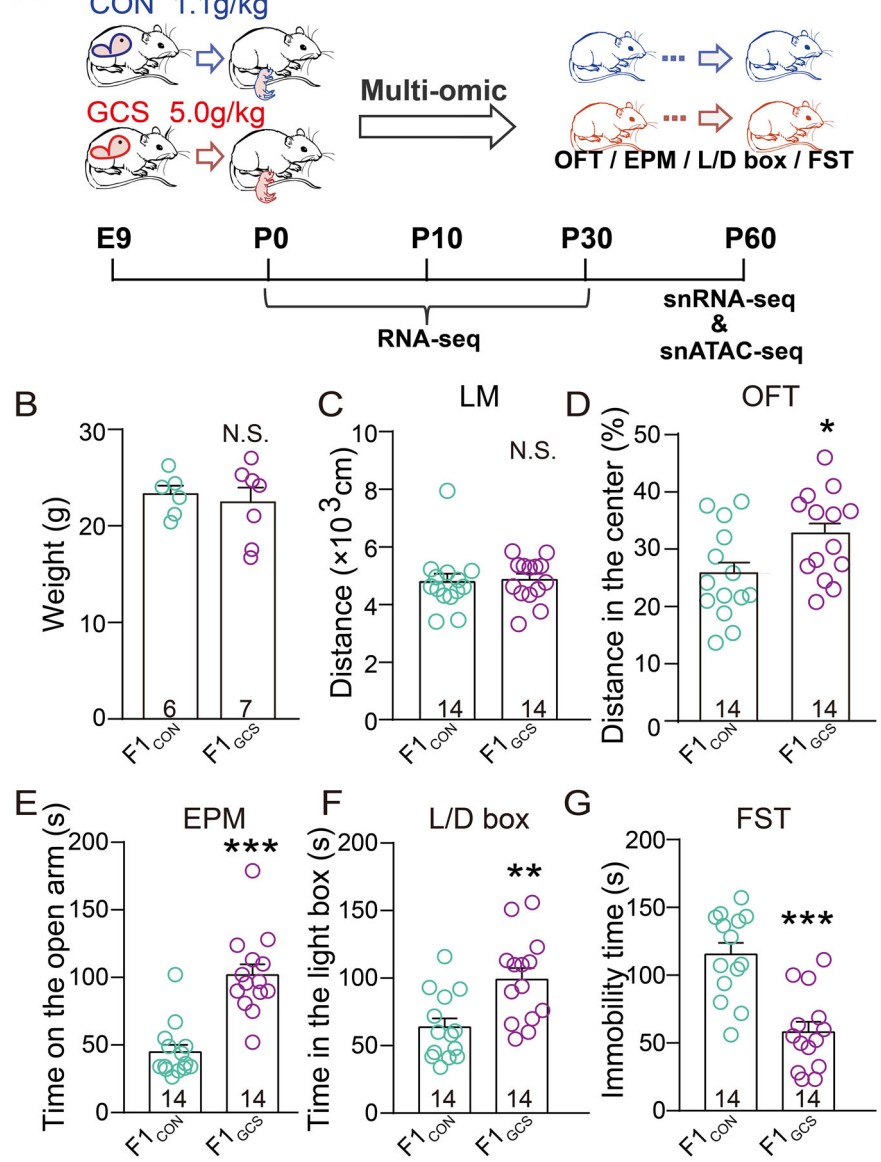

**Fig. 1 | GCS promotes anxiolytic and anti-depressive-like responses to acute stress in male offspring. A** Experimental design: Pregnant mice received choline chloride (CON: 1.1 g/kg; GCS: 5.0 g/kg) on embryonic day 9 (E9). F1 male offspring were analyzed at postnatal day (P) 0, P10, P30, and P60. RNA-Seq was performed on 3 hippocampi/group at P0, P10, and P30. At P60, snRNA-Seq and snATAC-Seq were performed on 5 hippocampi/group. The behavioral tests were conducted when the animals reached 2 months of age. All elements in this panel are original and were created by the authors. **B** The body weight of F1$_{GCS}$ showed no significant difference from that of F1$_{CON}$ ($n = 6$ mice for F1$_{CON}$ and $n = 7$ for F1$_{GCS}$). **C** There is no difference in locomotor activity (LM) over a 30-min period between F1$_{GCS}$ and F1$_{CON}$ male mice ($n = 14$). **D** F1$_{GCS}$ male mice travelled longer distances in the center area of the open field arena in the open field test (OFT) ($n = 14$; Cohen's d = 0.895). **E, F** F1$_{GCS}$ male mice spent more time in both the open arm of the elevated plus maze (EPM) test ($n = 14$; Cohen's d = 2.258) and the light side of the light-dark box (L/D box) test ($n = 14$; Cohen's d = 1.243). **G** F1$_{GCS}$ male mice displayed significantly lower immobility than F1$_{CON}$ male mice in the forced swim test (FST) ($n = 14$; Cohen's d = 1.916). (* $p < 0.05$; ** $p < 0.01$; *** $p < 0.001$). Data are represented as means ± SEMs.

male offspring of GCS, defined as $F1_{GCS}$, showed normal body weight ($p = 0.655$, Fig. 1B) and locomotor activity ($p = 0.285$, Mann–Whitney U test, Fig. 1C), suggesting that they achieved normal developmental milestones. In the open field test (OFT), $F1_{GCS}$ displayed higher locomotion activity in the center area of the open field arena than the offspring of CON ($F1_{CON}$) ($p = 0.026$, unpaired $t$ test, Fig. 1D), suggesting an anxiolytic effect of GCS. Consistently, the $F1_{GCS}$ male mice spent more time in the open arm of the elevated plus maze (EPM) ($p < 0.001$, Mann–Whitney U test, Fig. 1E) and the light side of the light-dark (L/D) box test ($p = 0.003$, Fig. 1F), suggesting that GCS alleviated anxiety-like behavior in F1 offspring. In a forced swim test (FST), which reflects depression-like behavioral responses to acute stress, $F1_{GCS}$ male mice showed significantly lower immobility than $F1_{CON}$ male mice ($p < 0.001$, Fig. 1G), suggesting the anti-depression effect. Notably, calculation of Cohen's d for the aforementioned behavioral tests demonstrated that 14 mice per group provided sufficient statistical power to detect meaningful intergroup differences (Fig. 1D–G).

## Multi-omic profiling reveals the comprehensive roles of GCS in diverse aspects of neuronal functions

To examine the transcriptional alterations and chromatin accessibility modulated by GCS at the single-nucleus level, the hippocampi of the male $F1_{GCS}$ and $F1_{CON}$ were collected at 2 months of age and their nuclei were extracted for single-nucleus RNA Sequencing (snRNA-Seq) and single-nucleus Assay for Transposase-Accessible Chromatin Sequencing (snATAC-Seq) analysis utilizing the DNBelab C4 platform (Fig. 1A)[34,35]. Following quality control, we retained a total of 54,859 nuclei (22,090 from $F1_{CON}$ hippocampus and 32,769 from $F1_{GCS}$ hippocampus) for snRNA-Seq analysis and 55,248 nuclei (34,523 from $F1_{CON}$ hippocampi and 20,725 from $F1_{GCS}$ hippocampi) for snATAC-Seq analysis (Supplementary data 1). Uniform Manifold Approximation and Projection (UMAP) dimensionality reduction was employed to identify distinct cell-type clusters in the snRNA-Seq and snATAC-Seq data from $F1_{CON}$ and $F1_{GCS}$ hippocampi[36] (Fig. 2A, B). Based on the gene expression patterns and activity characteristics of marker genes, the predominant cell types in the hippocampus, such as mature granule neuron, immature granule neuron, pyramidal neuron, oligodendrocyte, and astrocyte, were consistently identified via the analysis of both snRNA-Seq and snATAC-Seq data[25–30] (Fig. 2C–E, Figure S1A, B, Supplementary data 2).

In the $F1_{GCS}$ hippocampi, the numbers of differentially expressed genes (DEGs) and differentially accessible genes (DAGs) across distinct cell types identified by snRNA-Seq, snATAC-Seq, and shared between both sequencing approaches are shown in Figure S1C–E (Supplementary data 3). To determine whether the overlap between DEGs and DAGs in Fig. S1E exceeds random expectation, hypergeometric and Fisher's exact tests were conducted for both up- and down-regulated features in all cell types. No statistically significant enrichment was observed in any of the analyzed cell type (adjusted $p > 0.05$), suggesting that intermediate regulators, such as transcription factors or post-translational modifications, may mediate the chromatin accessibility and gene expression changes induced by gestational choline supplementation.

The snRNA-Seq analysis identified a total of 831 DEGs, comprising 568 up-regulated and 263 down-regulated ones (Supplementary data 3). Functional enrichment analysis of differentially up-regulated DEGs in the snRNA-Seq data were significantly enriched in the function of neural projection, synaptic organization, membrane potential regulation, cAMP signaling, and behavior in the $F1_{GCS}$ hippocampi ($p < 0.01$, Fig. 2F). In contrast, the functions of differentially down-regulated DEGs were related to energy metabolism such as ATP metabolism and canonical glycolysis (Fig. 2F). Meanwhile, the snATAC-Seq analysis identified 1,598 DAGs, including 777 up-regulated and 821 down-regulated ones (Supplementary data 3). Functional enrichment analysis of differentially up-regulated DAGs in the snATAC-Seq data were significantly enriched in the function of neural projection, synaptic organization, cell-cell adhesion, and glutamatergic synaptic transmission in the $F1_{GCS}$ hippocampi ($p < 0.01$, Fig. 2G), whereas the functions of differentially down-regulated DAGs were related to

energy metabolism such as RNA metabolism and protein ubiquitination (Fig. 2G). Notably, a total of 62 up-regulated and 17 down-regulated DEGs or DAGs were commonly identified in both the snRNA-Seq and snATAC-Seq datasets (Supplementary data 3), and their functions were enriched in synapse organization, synaptic transmission, regulation of membrane potential, glutamate receptor signaling (up-regulated) and energy metabolism such as cytoplasmic ribosomal proteins (down-regulated) (Fig. 2H). These results supported the extensive role of GCS in neuronal development and plasticity, as well as in neural protection.

## Significant differential expression and chromatin accessibility in granule neurons of $F1_{GCS}$ hippocampus

Notably, we observed a significantly increased proportion of immature granule neurons (for snRNA: adjusted $p = 0.021$; for snATAC: adjusted $p = 0.003$, unpaired $t$ test) and a slightly decreased proportion of mature granule neurons (for snRNA: adjusted $p = 0.164$; for snATAC: adjusted $p = 0.020$, unpaired $t$ test) from the $F1_{GCS}$ hippocampus (Fig. 3A, Supplementary data 1), suggesting potential alterations in the developmental trajectory or functional state of granule neurons induced by GCS. Compared with the $F1_{CON}$ group, qRT-PCR validation showed that the mRNA levels of immature granule neuron marker genes, such as *Dcx* and *Gda*, were significantly elevated in the hippocampi of $F1_{GCS}$ male mice at both P0 and adulthood, whereas the expression of the mature granule neuron marker gene *Calb1* was significantly reduced at P0 but showed no statistically significant difference at P60 (Fig. 3B, C). Consistently, DCX protein levels were significantly increased in the hippocampi of $F1_{GCS}$ male mice at both P0 and adulthood, as shown by Western blotting (Fig. 3D, E, Figure S2).

To identify the altered gene expression patterns, epigenetic modifications, or changes in cellular signaling pathways that may contribute to the behavioral phenotypes, an integrated analysis of gene expression and chromatin accessibility at the single-nucleus level were employed. In the immature granule neurons of the $F1_{GCS}$ hippocampus, the snRNA-Seq analysis showed that up-regulated DEGs were associated with functions related to synapse organization, axo-dendritic transport, axonogenesis, and glutamate receptor signaling pathway (Fig. 3F). snATAC-Seq further demonstrated that functional enrichment of DAGs mirrored that observed in the gene expression analysis. Specifically, DAGs in immature granule neurons were predominantly enriched in processes related to neuronal connections and signal transmission, including neuronal projection morphogenesis, synapse organization, as well as the cell-cell adhesion and ECM-receptor interaction pathways (Fig. 3G). These processes are critical for the migration and positioning of immature neurons. Notably, while their proportion decreased, the functions of up-regulated DEGs in mature granule neurons were enriched in key neuronal processes linked to synaptic transmission, synapse organization, and cholinergic synapses; meanwhile, the up-regulated DAGs in them were involved in functional maintenance and systemic integration, including calcium signaling, GPCR signaling, positive regulation of the MAPK cascade, and locomotion (Fig. 3F, G). These results suggested that GCS may attenuate the anxiety- and depression-like behaviors in offspring through regulating granule neuron development and function.

## GCS enhances intercellular communication in immature granule neurons

To investigate how GCS influences the function of granule neurons, we assessed whether intercellular communication involving immature and mature granule neurons were regulated by GCS, using the CellChat (v.2) software[37]. Our analysis indicated that secreted signaling (such as growth factors and cytokines), ECM-receptors in cell adhesion and extracellular matrix interactions, cell-cell contact (involves junctional complexes and adhesion molecules), and non-protein signaling (includes various chemical messengers) accounted for 37.9%, 12.9%, 19.6%, and 29.7% of the total neural interactions, respectively (Fig. S3A). Compared to the interactions in the hippocampus of $F1_{CON}$, the total interaction strength, rather than its quantity, was significantly enhanced in $F1_{GCS}$ ones ($p < 2.2e-16$, Wilcoxon

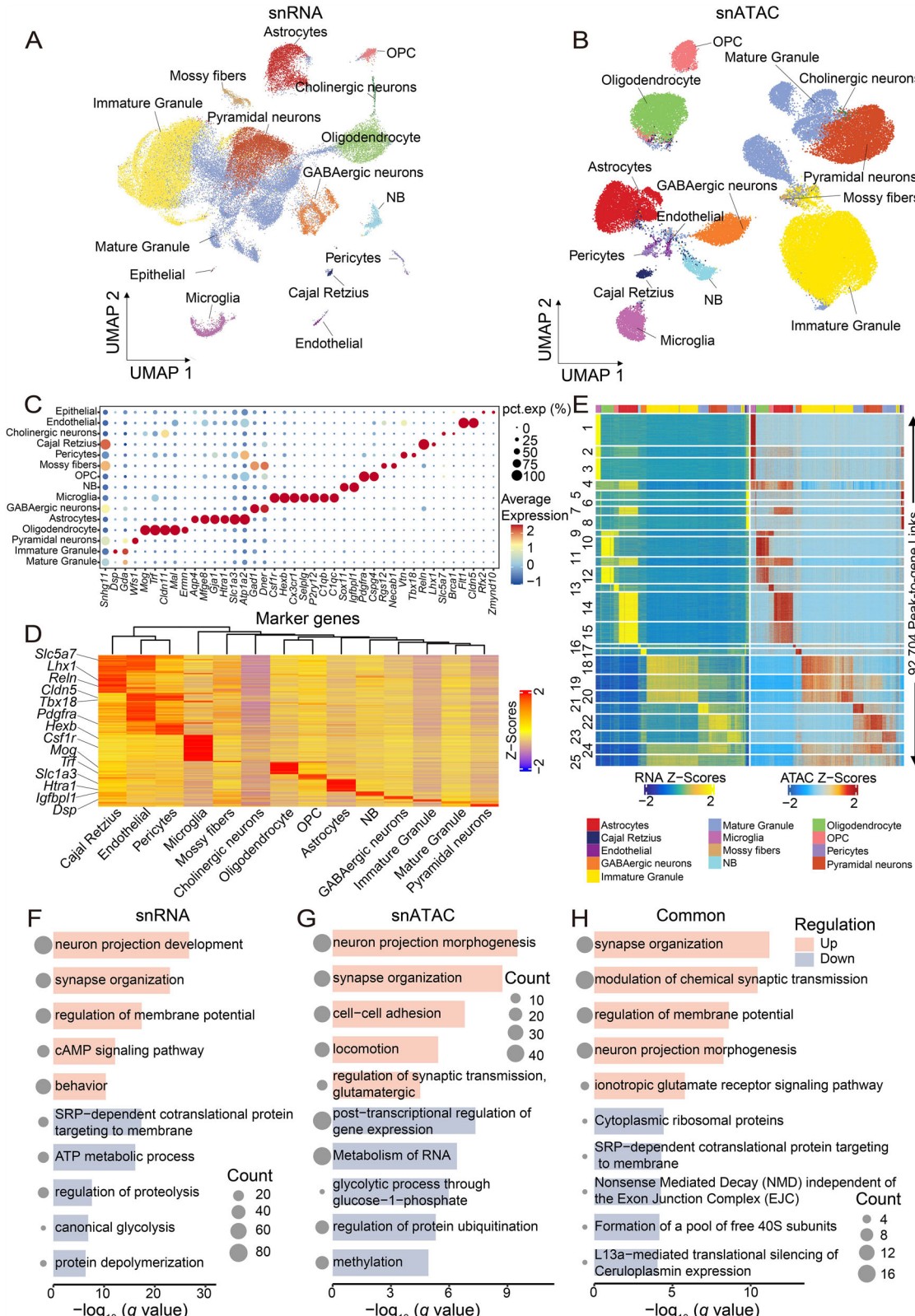

**Fig. 2 | Multimodal atlas of the hippocampi of F1$_{CON}$ and F1$_{GCS}$ male mice at P60.**
**A** UMAP analysis of 54,859 snRNA-Seq profiles identified 15 mouse hippocampal cell populations. Dots are colored by cell type. **B** Based on gene activity scores of established marker genes, UMAP analysis of 55,248 snATAC-Seq profiles identified 14 mouse hippocampal cell populations at the chromatin level. Dots are colored by cell type. **C** Dot plot showing the expression levels of representative marker genes across annotated cell types. The color scale indicates average gene expression, and dot size denotes the percentage of cells expressing these marker genes within each subpopulation. **D** Heatmap of marker genes (gene scores) from snATAC-Seq for the indicated cell types. The color scale represents the relative Z-score of each marker gene. **E** Multi-modal feature map of the mouse hippocampus, including snRNA gene expression and relative snATAC gene activity in embedded snATAC cells. **F–H** Pathways enriched by differentially expressed (F) /chromatin-accessible (G) and common (H) genes that are up-regulated (upper) or down-regulated (lower) in F1$_{GCS}$ from snRNA-Seq, snATAC-Seq, and both omics datasets. Circle size represents the number of genes enriched in each pathway.

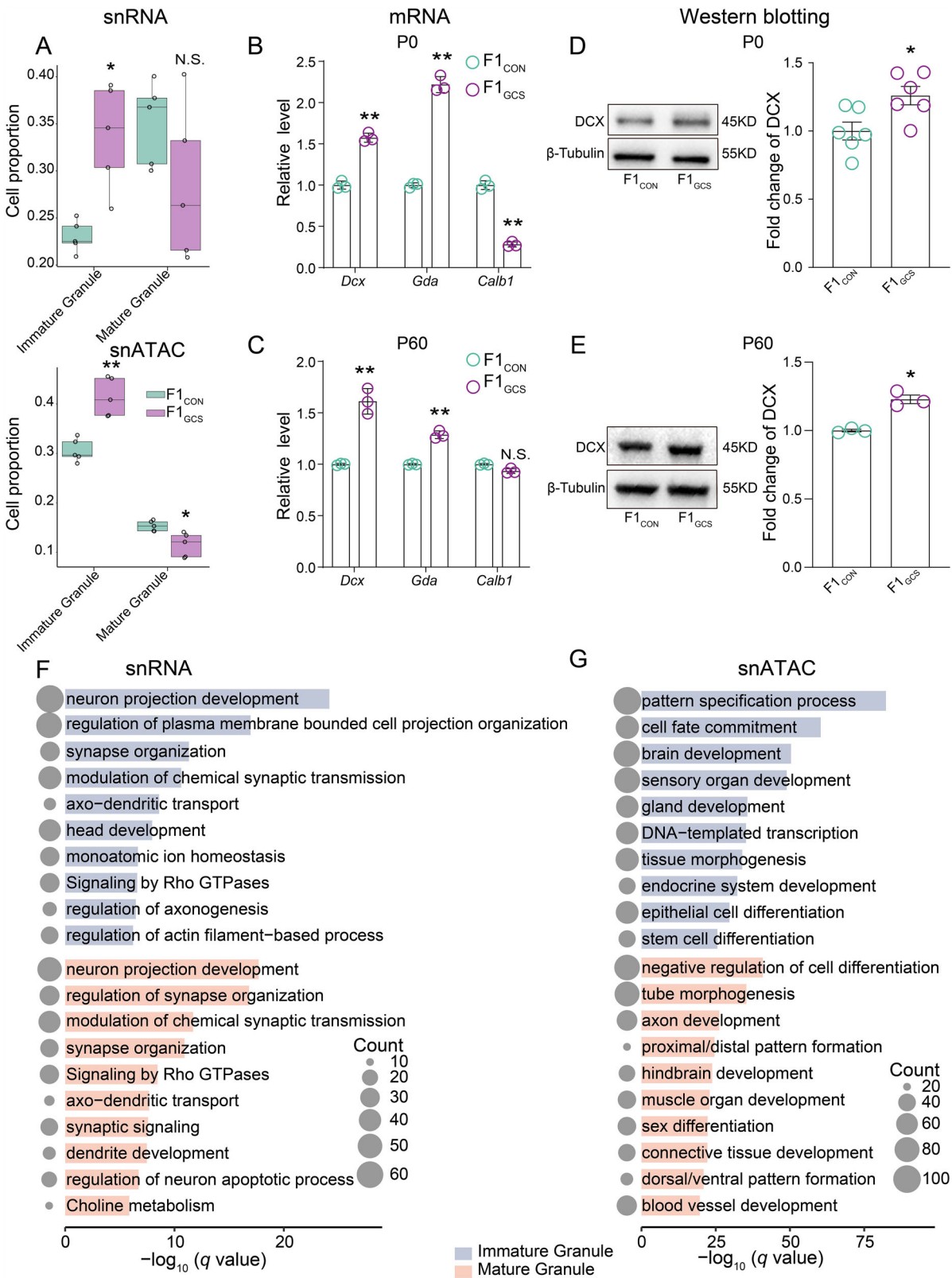

test), particularly for cell-cell contact ($p < 2.2e-16$, Wilcoxon test) and non-protein signaling ($p < 2.2e-16$, Wilcoxon test) (Fig. 4A). Notably, immature granule neurons exhibited the most pronounced and significantly increases in interaction strength, whether occurring within these cells or with other cell types in $F1_{GCS}$ hippocampus ($p = 2.791e-41$, unpaired $t$ test, Fig. 4B, C and Fig. S3B). In immature granule neurons, enhanced interactions (Groups 1, 2) were enriched in cell adhesion (e.g., NRXN, PTPR), axonogenesis and synaptogenesis pathways (e.g., SEMA5/6, UNC5, PTPR, and NRXN), and excitatory neurotransmitter glutamate, all of which are directly related to emotion-like behaviors such as anxiety and depression. Dysfunction in these molecules has been linked to psychiatric disorders[38–41]. In contrast, weakened interactions (Groups 3–4) in immature granule neurons were

**Fig. 3 | Proportion changes and functional characteristics of immature and mature granule neurons in the F1$_{CON}$ and F1$_{GCS}$ mice hippocampus. A** Cell proportion analysis of the percentage of each granule cell population (immature and mature granule neurons) in snRNA-Seq (upper) and snATAC-Seq (lower) data from individual samples. Colors represent groups, and asterisks indicate significance levels. For both snRNA-Seq and snATAC-Seq, 5 F1$_{CON}$ and 5 F1$_{GCS}$ mouse samples were used. In box plots, boxes denote upper/lower quartiles, and lines represent medians. **B, C** The levels of marker genes for immature (*Dcx* and *Gda*) and mature (*Calb1*) granule neurons including *Dcx* (P0: $p = 0.005$, P60: $p = 0.005$), *Gda* (P0: $p = 0.005$; P60: $p = 0.005$), and *Calb1* (P0: $p = 0.005$; P60: $p = 0.152$) in the hippocampus of F1$_{CON}$ and F1$_{GCS}$ mice at P0 (B) and adulthood (C). $n = 3$ per group, Mann–Whitney U test. **D, E** The protein levels of DCX in the hippocampus of F1$_{CON}$ and F1$_{GCS}$ mice at P0 (D, $n = 6$; $p = 0.011$) and adulthood (E, $n = 3$; $p < 0.05$). Mann–Whitney U test. **F, G** Pathways enriched by differentially expressed (snRNA-Seq, F) and differentially accessible (snATAC-Seq, G) genes in immature (upper) and mature (lower) granule neurons that are up-regulated. Circle size corresponds to the number of genes enriched in each pathway. * (adjusted) $p < 0.05$; ** (adjusted) $p < 0.01$. Data are represented as means ± SEMs.

associated with metabolic pathways (e.g., ApoE-linked lipid metabolism[42]) and immune pathways (e.g., MIF, CXCL) (Fig. 4D).

The cellular communication of glutamate neurotransmitters that are directly related to emotion regulation was significantly enhanced in immature granule neurons ($p < 0.001$, unpaired *t* test, Fig. 4E and Figure S3C, D). The key ligand-receptor molecules in this signaling, *Gria1* and *Gria2*, were significantly up-regulated in both their expression levels and chromatin accessibility in immature granule neurons of the F1$_{GCS}$ hippocampus (Fig. 4F, G). The increased mRNA levels of *Gria1* and *Gria2* in the hippocampus of F1$_{GCS}$ were validated by qRT-PCR at both P0 and adulthood (Fig. 4F). Additionally, snRNA-Seq analysis and qRT-PCR validation showed an upregulation of the glutamatergic signaling gene *Grik5* in the hippocampus of the F1$_{GCS}$ mice at P60 (Figure S3E). Other components in glutamates signaling, including *Slc1a1*, *Grin2a*, *Grm1*, *Grm2*, and others, were increased in the hippocampus of the F1$_{GCS}$ mice at P0 (Fig. S4A–C).

Moreover, the cellular communication of MIF, a macrophage migration inhibitory factor, was significantly weakened in immature granule neurons in F1$_{GCS}$ hippocampus as compared to F1$_{CON}$ ones (Fig. S3D). Consistently, *Mif* showed a reduced expression level and chromatin accessibility in the immature granule neurons of F1$_{GCS}$ hippocampus (Fig. 4H, I). The decreased mRNA levels of *Mif* in the F1$_{GCS}$ hippocampus were validated by qRT-PCR at P0 (Fig. 4H). Consistently, several receptors that bind to *Mif*, including *Ackr3*, *Cd74*, *Cxcr2*, and *Cxcr4*, were significantly down-regulated in the hippocampus of F1$_{GCS}$ mice at P0 (Fig. S4D). These findings suggest that GCS may modulate neuron development and functions through the regulation of cell adhesion, axonogenesis and neurotransmission, as well as via effects on the immune system.

## GCS exerts stage-specific regulation during hippocampal development

To investigate the impact of GCS on early hippocampal development, we performed bulk RNA-Sequencing (RNA-Seq) of the hippocampi of F1$_{CON}$ and F1$_{GCS}$ male mice at P0, P10, and P30 ($n = 3$). Functional enrichment analysis of DEGs in the hippocampi of F1$_{GCS}$ male mice revealed distinct, age-specific profiles at each developmental stage. At P0, upregulated pathways were associated with neural morphogenesis and pattern specification, while downregulated pathways were related to inflammatory response and developmental homeostasis (Fig. S5A). At P10, upregulated DEGs were enriched in neuronal maturation processes, such as axon development and projection morphogenesis, whereas downregulated pathways were involved extracellular matrix organization, ion homeostasis, and antioxidant response (Fig. S5B). By P30, upregulated functions were associated with immune regulation and neural signaling, including innate immune response and serotonin secretion modulation, while downregulated terms were enriched in cytoskeletal dynamics and protein ubiquitination (Fig. S5C). The overlap among up-regulated, down-regulated, and all differentially expressed genes in the hippocampus of F1$_{GCS}$ mice across different stages were shown in Fig. S5D–G. Of note, *Lpar1* and *Ttr* showed persistent differential expression from P10 to P60 (Fig. S5G). Moreover, *Lpar1* showed increased chromatin accessibility in mature granule neurons and astrocytes at P60 (Figure S5H). These results suggest that prenatal nutritional interventions may modulate brain function and behavior in a developmentally stage-specific manner.

In particular, transcriptomic profiling showed that genes associated with the cholinergic signaling were expressed at higher levels at P0 (Fig. 5A),

suggesting that the perinatal period represents a critical window for choline supplementation. Compared to F1$_{CON}$ mice, the hippocampi of F1$_{GCS}$ male mice exhibited significantly higher levels of these genes, such as *Chrna3*, *Creb3l2*, *Itpr3*, and *Pla2g4b*, which were validated by qRT-PCR at P0 (Fig. 5A, B). This observation also suggests the effects of GCS on cholinergic signaling. Additionally, genes related to glutamatergic synapses showed peak expression at P30 (Fig. 5A–C), indicative of stage-specific synaptic maturation.

Bulk differential expression profiling consistently highlighted a considerably downregulation of *Mif* and a broader set of genes in MIF-related pathways in F1$_{GCS}$ mice relative to controls (Fig. 5A, D). Pathway enrichment analysis further revealed that genes involved in tyrosine and phenylalanine metabolic pathways—specifically those modulated by the *Mif* gene—were broadly expressed throughout postnatal hippocampal development in mice (Fig. 5A). Notably, these differentially expressed genes were significantly enriched in glucocorticoid response pathways ($p < 0.01$, Fig. 5A), suggesting a regulatory axis whereby GCS modulates glucocorticoid signaling through *Mif*-dependent mechanisms, potentially influencing offspring emotional phenotypes. Collectively, these findings underscore the stage-specific effects of GCS on early neural circuit assembly.

## Major depressive disorders (MDD) and anxiety-related genes in F1$_{GCS}$ hippocampus showed altered correlation with the diseases during brain development

Given the essential functions of the hippocampal neurons in various neurological and psychiatric disorders[43–47], we integrated snRNA-Seq and Genome-Wide Association Study (GWAS) data and performed a single-cell Disease Relevance Score (scDRS) analysis to identify GCS-modulated diseases at the single-nucleus level[48]. Our results indicated that diverse types of neurons in the F1$_{GCS}$ hippocampus were substantially correlated with a range of brain-related diseases or phenotypes, including MDD and anxiety (Fig. S6A and Supplementary data 4). We further assessed the associations between MDD and anxiety-related gene expression and scDRS in the hippocampus of F1$_{GCS}$ and F1$_{CON}$ using Pearson correlation analysis. As a result, the correlation coefficients between these genes and MDD or anxiety were decreased across all hippocampal cell types in F1$_{GCS}$ as compared to those in the F1$_{CON}$ mice (Fig. S6B). In particular, the correlation coefficients between *Mif*, *Gria1*, and *Gria2* genes and MDD or anxiety exhibited a consistent downward trend across all hippocampal cell types in the F1$_{GCS}$ mice (Fig. S6C).

Notably, immature granule neurons showed stronger correlation with neuropsychiatric diseases, including MDD and anxiety (Fig. 6A). The correlation with MDD and anxiety were significantly lower in immature granule neurons of F1$_{GCS}$ compared to those in F1$_{CON}$ (Fig. 6B). For the top 300 genes that exhibited reduced correlation with MDD and anxiety in immature granule neurons (Fig. 6C), pathway enrichment analysis identified terms such as "regulation of synapse organization", "synaptic signaling", "protein-protein interactions at synapses", and "phosphatidylinositol metabolic process" (Fig. 6D). Additionally, genes linked to MDD also showed significant enrichment in the phosphatidylinositol metabolic process and regulation of dephosphorylation pathways ($p < 0.01$, Fig. 6D), representing potential metabolism-related pathways. These genes exhibited stage-specific expression in immature granule neurons (Fig. 6E). These findings consistently supported the stage-specific role of GCS in regulating neural function and communication involved in alleviating anxiety/depression-like behaviors.

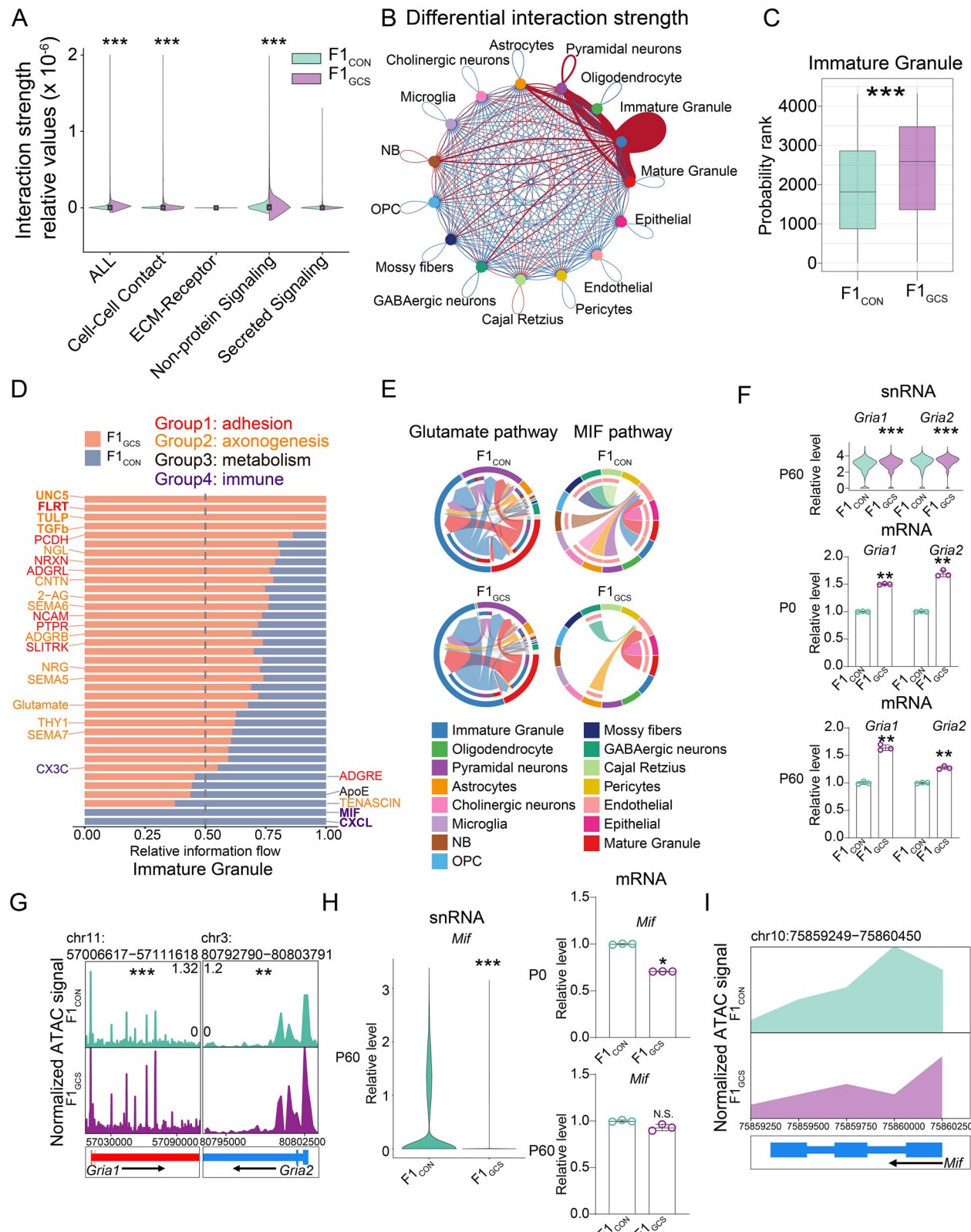

## Discussion

Emerging evidence has established that maternal nutrition profoundly affects offspring development[1,7]. Choline, an essential nutrient, is known to support neurodevelopment and cognitive function. However, the cellular and molecular mechanisms through which GCS influences complex behavioral outcomes at single-cell level remain to be fully elucidated.

Granule neuron maturation in the hippocampus is closely linked to neural circuit plasticity, and associated functional impairments are commonly observed in emotional disorders such as anxiety and depression. Whether GCS modulates granule neuron development remains to be determined. In this study, our integrated analysis of bulk transcriptomic, single-nucleus transcriptomic, and single-nucleus ATAC data revealed a significant impact

**Fig. 4 | Cell communication analysis of F1$_{CON}$ and F1$_{GCS}$ mice hippocampus at P60. A** Bar chart comparing the cell communication strength of five types (ALL, Cell-Cell Contact, ECM-Receptor Signaling, Non-protein Signaling, and Secreted Signaling) between F1$_{CON}$ (green) and F1$_{GCS}$ (purple) groups. The y-axis shows relative interaction strength; significant differences are indicated by asterisks. **B** Circos plot showing changes in intercellular and intracellular communication strength among 15 hippocampal cell types in P60 F1$_{CON}$ and F1$_{GCS}$ mice. In the color bar, red indicates enhanced and blue indicates weakened signaling in F1$_{GCS}$ relative to F1$_{CON}$. **C** Cell communication probability rank of the immature granule neurons from F1$_{CON}$ (green) and F1$_{GCS}$ (purple) mice. Colors represent groups, and the asterisks with different quantities indicate significance levels. For box plots, the boxes represent the upper and lower quartiles, and the horizontal lines within represent the medians. **D** Sum of relative interaction strength differences (relative information flow) across all interaction pairs in F1$_{CON}$ (purple) and F1$_{GCS}$ (orange) groups. Interactions are categorized as adhesion (red), axonogenesis (orange), metabolism (brown), and immune (purple). **E** Glutamate (left) and MIF (right, probability = 1.030e-10 in F1$_{CON}$ and probability = 0 in F1$_{GCS}$) signaling networks in F1$_{CON}$ (upper) and F1$_{GCS}$ (lower); nodes represent cell types, and chords denote interactions, with width proportional to interaction strength. **F** Top: Violin plots of *Gria1* (adjusted $p$ = 1.212e-37, log$_2$FC = 0.163) and *Gria2* (adjusted $p$ = 6.393e-37, log$_2$FC = 0.161) relative expression in immature granule neurons between F1$_{CON}$ and F1$_{GCS}$. Middle–Bottom, the levels of *Gria1* (P0: $p$ = 0.005; P60: $p$ = 0.005) and *Gria2* (P0: $p$ = 0.005; P60: $p$ = 0.005) in the hippocampus of F1$_{GCS}$ mice at P0 and P60 validated using the qRT-PCR approach. **G** Genome browser tracks showing single-nucleus chromatin accessibility at the *Gria1* (FDR = 9.342e-13, log$_2$FC = 0.424) and *Gria2* (FDR = 0.007, log$_2$FC = 0.236) loci in immature granule neurons between F1$_{CON}$ and F1$_{GCS}$. **H** Left: Violin plot of *Mif* (FDR = 1.401e-05, log$_2$FC = -0.128) relative expression in immature granule neurons between F1$_{CON}$ and F1$_{GCS}$ mice. Right: the levels of *Mif* in the hippocampus of F1$_{GCS}$ mice at P0 ($p$ = 0.010) and P60 ($p$ = 0.076) validated using the qRT-PCR approach. **I** Genome browser tracks showing single-nucleus chromatin accessibility at the *Mif* locus (FDR = 0.938, log$_2$FC = -0.481) in immature granule neurons between F1$_{CON}$ and F1$_{GCS}$. Mann–Whitney U test was performed for qRT-PCR analyses. * (adjusted) $p$ < 0.05; ** (adjusted) $p$ < 0.01; *** (adjusted) $p$ < 0.001. Data are represented as means ± SEMs.

of GCS on hippocampal development and complex behaviors. In particular, alterations in differential gene expression, chromatin accessibility, cellular communications in immature granule neurons offer insights into the potential mechanisms by which GCS may influence the emotional states of offspring. Bulk transcriptome analysis across multiple developmental stages further suggests that GCS modulates neuronal development and biological events during critical developmental periods, such as the cholinergic signaling pathway around P0, the *Mif*-associated metabolic network around P10, and the glutamatergic neurotransmission system around P30. The decreased transcriptional correlation between the immature granule neurons and anxiety and depression further provide cellular evidence for the observed behavioral changes. Thus, our study provides multifaceted evidence to elucidate the cell-type-specific effects of GCS on offspring emotional states.

The association between granule cell dysfunction and anxiety and depressive disorders has been well established[49]. For example, patients with MDD often exhibit a reduction in the hippocampal volume, particularly in the dentate gyrus region, where granule neurons mainly reside[20,21]. Antidepressant drugs have been shown to enhance neurogenesis in the adult dentate gyrus by a mechanism involving the transformation of mature granule neurons into a more immature state, which is manifested as an increase in neuronal excitability and a down-regulation of the expression of mature granule cell markers[50,51]. Immature granule neurons exhibit enhanced synaptic plasticity, enabling the formation of new connections more efficiently, while the suppression of mature granule neurons activity reflects a shift to a more plastic and adaptable cellular state, thereby improving resilience to chronic stress[24,52]. Consistently, we found increased proportion of immature granule neurons but decreased proportion of mature granule neurons in the hippocampus of F1$_{GCS}$ mice. These observations suggest that GCS may regulate granule neuron development to foster a neurogenesis-promoting environment, ultimately contributing to anti-anxiety and -depressive effects.

Multi-omics sequencing in this study revealed that GCS regulates hippocampal granule neuron development and modulates cellular communication within the hippocampal microenvironment. Although standardized sampling of intact hippocampal tissue was employed to control for volume variability, the resolution of subregion-specific effects remains limited. From a neural circuitry perspective, the hippocampus does not function in isolation but as part of a distributed network involving the amygdala and hypothalamus[53,54]. For example, human neuroimaging studies provide evidence of dynamic functional interactions between the hippocampus and the amygdala during emotional processing[55]. Thus, the observed GCS-induced alterations in intra-hippocampal cellular communication may influence emotional states by modulating hippocampal outputs to these broader circuits. Future studies applying spatial transcriptomics or subregion-specific laser microdissection could more precisely delineate the regulatory roles of choline within distinct hippocampal subregions.

At the cellular level, a broad enhancement of communication pathways involving axon guidance and anxiety- or depression-like behaviors was observed in the immature granule neurons of the F1$_{GCS}$ hippocampus. For example, UNC5, an axon guidance regulator whose dysfunction is associated with adolescence psychiatric disorders[41], is specifically up-regulated in these neurons. Several members of the SEMA family, including SEMA5, SEMA6, and SEMA7, are also up-regulated in immature granule neurons of F1$_{GCS}$ hippocampus, and these genes are linked to emotional regulation, metabolism processes, and inflammatory responses[40]. Notably, deficiency of SLIT and NTRK like protein-5 (*Slitrk5*) leads to obsessive-compulsive behaviors in mice, as evident by excessive self-grooming and increased anxiety-like behaviors[56]. Moreover, signals related to neurotransmitters (e.g., glutamate) and cell adhesion molecules (e.g., NRXN and ADGRL) are implicated in abnormal neurodevelopment. Neurexin (NRXN) dysfunction has been associated with multiple neurodevelopmental disorders[39], while ADGRL1 haploinsufficiency results in persistent developmental, neurological, and behavioral abnormalities in both mice and humans[57]. Thus, the upregulation of these signaling pathways in the F1$_{GCS}$ hippocampus may represent adaptive responses to support neurodevelopmental integrity and behavioral resilience. Notably, bulk transcriptomic analysis indicated that *Lpar1*, a DAG in mature granule neurons and astrocytes, exhibited differential expression beginning at P10 and persisting through P60 in the hippocampus of F1$_{GCS}$ mice, suggesting that GCS induced enduring epigenetic remodeling at this locus. Given its established roles in neuronal migration, myelination, and stress responsiveness[58], sustained upregulation of *Lpar1* may reflect a mechanism of transcriptional memory that contributes to the observed anxiolytic and antidepressant phenotypes in F1$_{GCS}$ mice.

Single-cell disease association analysis facilitates the investigation of correlations between cell profiles and diseases phenotypes at the single-cell resolution. Using single-cell disease relevance analysis (scDRS), we further assessed the association between the gene expression profiles of immature granule neurons in the hippocampus of F1$_{GCS}$ mice and various disease conditions. Our results showed that GCS effectively alleviated anxiety- and depression-like behaviors in F1 offspring, with reduced disease relevance observed in both the expression patterns and functional characteristics of associated genetic elements. These changes are closely associated with improved neuronal function and resilience to stress, highlighting the translational significance of GCS.

It is important to note that this study focused exclusively on male offspring, due to the potential confounding effects of hormonal fluctuations associated with the female estrous cycle, which can influence a range of behavioral experimental readouts[59,60]. However, considering the growing evidence demonstrating that intergenerational effects can exert sex-specific impacts on behavioral outcomes, gene expression profiles, and epigenetic

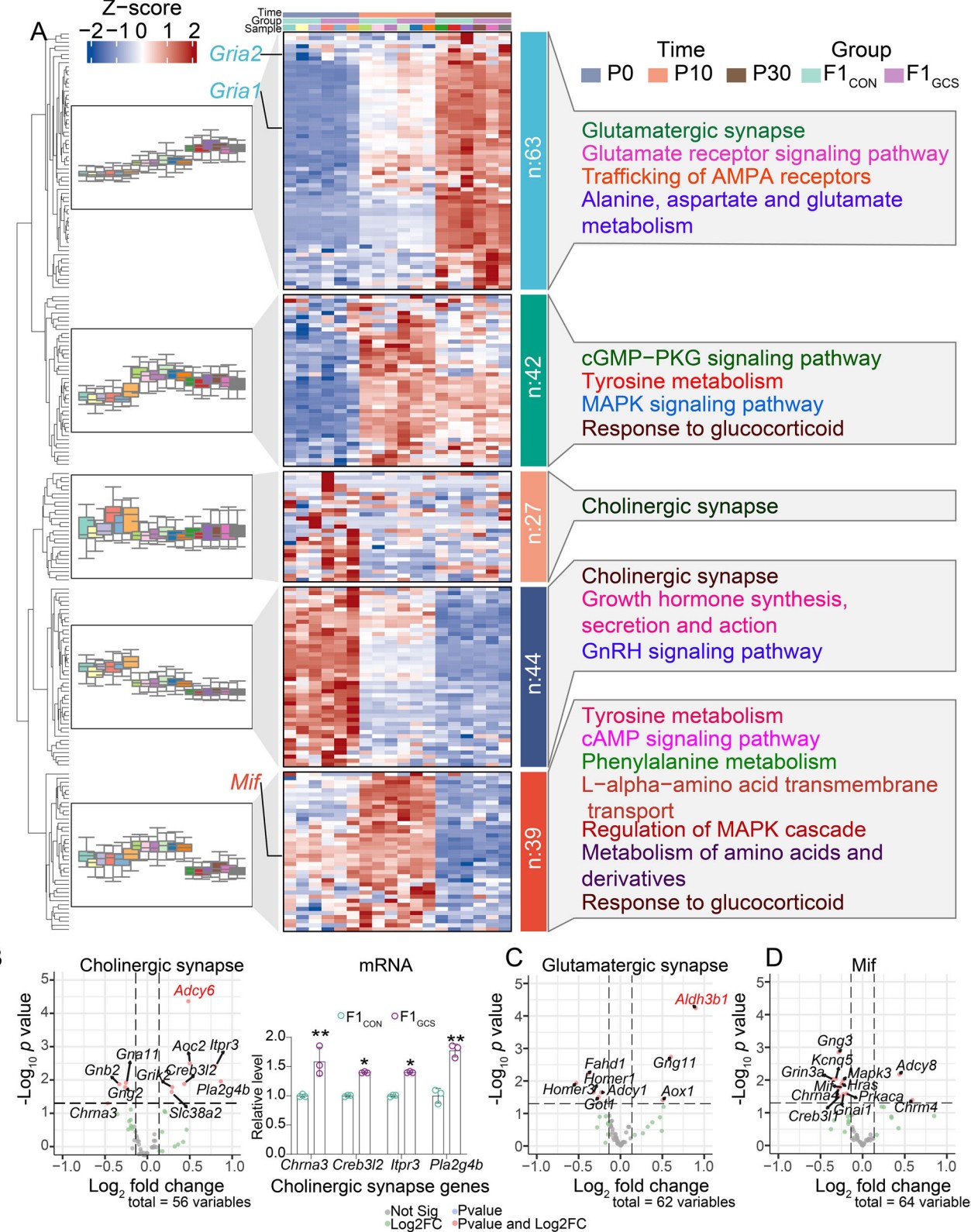

**Fig. 5 | Regulatory role of GCS during hippocampal development. A** Expression distributions of genes involved in glutamate-, MIF-, and choline-related pathways in the hippocampus of F1$_{CON}$ and F1$_{GCS}$ mice at P0, P10, and P30. The left boxplot shows trends in gene expression changes during hippocampal development; the heatmap presents relative gene expression levels (Z-score); the right panel displays the five clustered pathways. **B** Left: differential expression of genes involved in cholinergic synapse pathways in the hippocampus between F1$_{CON}$ and F1$_{GCS}$ mice at P0. Right: The expression levels of cholinergic signaling-related genes including

*Chrna3* ($p = 0.005$), *Creb3l2* ($p = 0.010$), *Itpr3* ($p = 0.010$), and *Pla2g4b* ($p = 0.005$) in the hippocampus of F1$_{CON}$ and F1$_{GCS}$ mice at P0 were validated by qRT-PCR. **C** Differential expression of genes related to glutamatergic signaling in the hippocampus between F1$_{CON}$ and F1$_{GCS}$ mice at P60. **D** Differential expression of genes associated with MIF-related pathways in the hippocampus between F1$_{CON}$ and F1$_{GCS}$ mice at P10. In the volcano plot of differential gene expression analysis, only the differentially expressed genes with adjusted $p < 0.05$ were displayed, which were highlighted in red. * $p < 0.05$; ** $p < 0.01$. Data are represented as means ± SEMs.

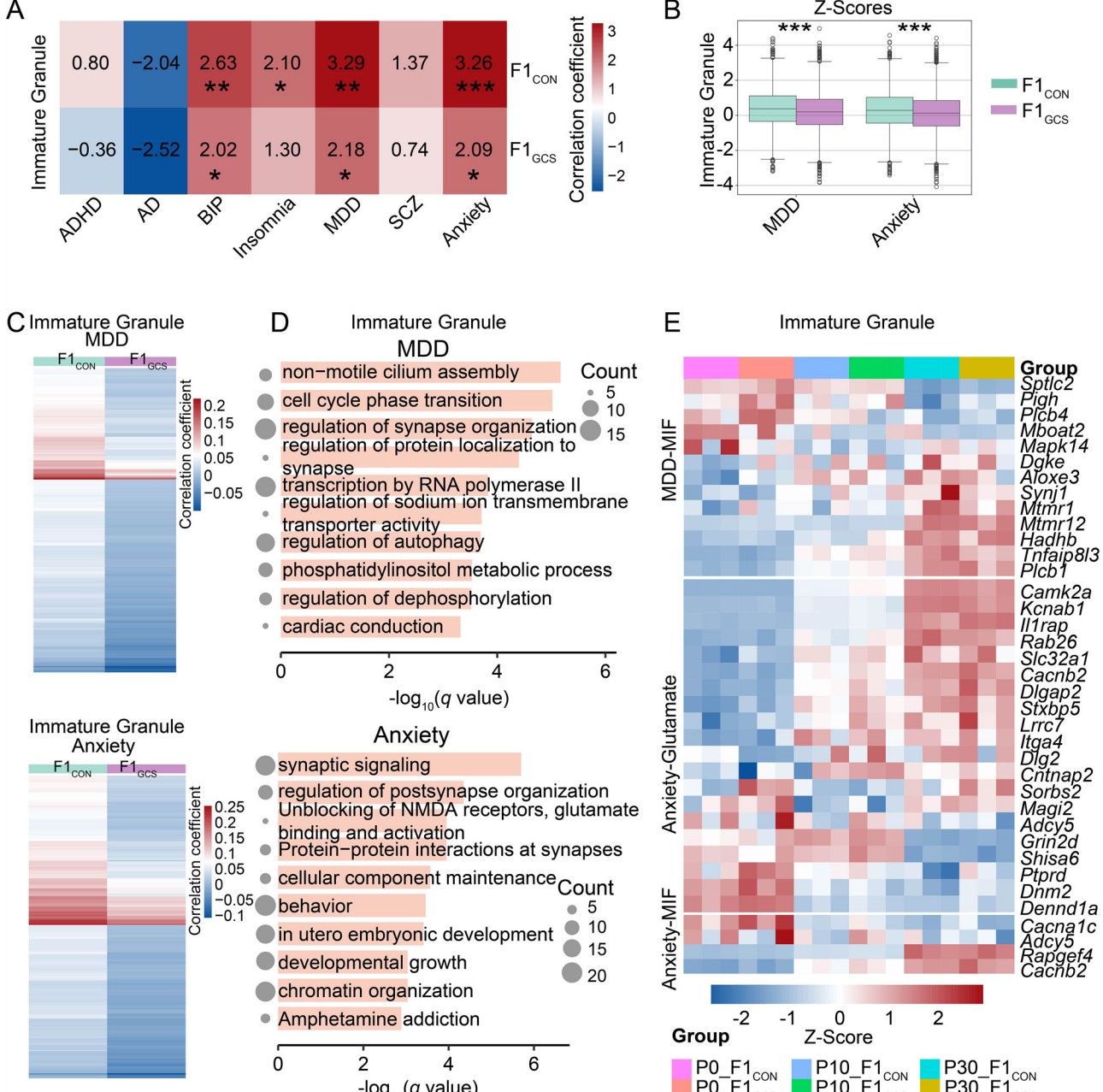

**Fig. 6 | GCS reduces correlation of immature granule neurons in mouse hippocampus with MDD and anxiety. A** scDRS correlation coefficients of immature granule neurons in F1_CON and F1_GCS mice, with respect to diseases including attention deficit hyperactivity disorder (ADHD), Alzheimer's disease (AD), bipolar disorder (BIP), insomnia, major depressive disorder (MDD), schizophrenia (SCZ), and anxiety. The color bar indicates correlation coefficients, and asterisks denote statistical significance. **B** Correlation analysis between individual cells in the immature granule neuron population of F1_CON (green) and F1_GCS (purple) mice and MDD/anxiety. Colors represent groups, and asterisks with different quantities indicate significance levels (unpaired *t* test, * *p* < 0.05; ** *p* < 0.01; *** *p* < 0.001). For box plots, boxes represent upper/lower quartiles, and lines inside represent medians. **C** Correlation coefficients in F1_CON and F1_GCS mice for the top 300 genes with the most significant reduction in correlation with MDD and anxiety in immature granule neurons of F1_GCS. **D** Functional enrichment results for the top 300 genes associated with MDD (upper) and anxiety (lower) in immature granule neurons. The size of circles corresponds to the number of genes enriched in each pathway. **E** Expression distribution of genes involved in glutamate- and MIF-related pathways associated with MDD and anxiety in hippocampal immature granule neurons of F1_CON and F1_GCS mice at P0, P10, and P30.

states[61–64], future investigations including female subjects will be essential to comprehensively understand how GCS influences emotional outcomes in both sexes.

In summary, this study demonstrates that GCS exerts a profound influence on hippocampal granule neuron development in mice. The upregulation of key pathways and processes such as glutamate signaling, neuron projection, and synapse organization, underscores the cell-type-specific effects on GCS-mediated neuroprotection. These findings not only elucidate the molecular and cellular pathways through which GCS modulates neural development and behavioral outcomes, but also suggest potential therapeutic strategies for the prevention of emotional disorders.

## Materials and Methods
### Mice and gestational Choline Supplementation (GCS) Program
Six-week-old male and female C57BL/6 J mice purchased from the Animal Center of Vital River Laboratories (Beijing, China) were mated in pairs and

kept under a normal 12:12 h light-dark cycles. All mice were given *ad libitum* access to standard chow and water, except during gestational stage as described below. From embryonic day 9 (E9) to the day of birth, half of the pregnant dams received a choline-supplemented diet (5 g/kg choline chloride; F0$_f$-GCS), while the remaining dams received a normal diet (1.1 g/kg choline chloride; F0$_f$-CON) in accordance with established protocols on choline supplementation across various mouse models[12,33]. These intervention period commenced after the completion of major neural tube closure events[65,66] and represented an important window for epigenetic reprogramming[1] and a key period for nervous system development[17]. Offspring derived from F0$_f$-GCS (namely F1$_{GCS}$) and F0$_f$-CON mice (namely F1$_{CON}$) were fed with a basal diet after birth and subjected to mood-like behavioral tests at 2 months old. To ensure statistical independence and account for biological variability, this study employed a split-experiment design with independent sampling across experimental modules. All dams were maintained under identical gestational conditions. Litter size varied between 7–10 pups, with a sex ratio close to 1:1. Only 1–2 male mouse per dam was used for each of the molecular and behavioral experiments and each individual offspring mouse was treated as an independent biological replicate in all subsequent analyses. Accordingly, a sufficient number of pregnant dams were used to ensure a final sample size of $n = 14$ mice per group for behavioral tests. All experiments were approved by the Animal Use Committee of the Institute of Zoology, Chinese Academy of Sciences, and in accordance with the guidelines of the National Institutes of Health (NIH).

## Mouse behavioral assays

Locomotor activity and open field test, forced swim test, elevated plus maze, as well as light-dark box test were performed according to the protocol previously published[67–70]. Briefly, the male F1$_{CON}$ and F1$_{GCS}$ mice were pretreated with gentle handling for 2–3 min each day for 4 consecutive days to eliminate their nervousness. Before each behavioral tests, the mice were allowed to habituate in the behavior testing room for 30 min. The behavior testing rooms were soundproof and big enough to keep a reasonable distance between mice and experimenter and the illumination level in the behavior room was maintained at 100 lux.

### Locomotor activity and open field test (OFT).
The locomotor activity and OFT were performed according to the protocol previously published[67–70]. Briefly, the equipment for the locomotor activity and OFT was a square arena (30 cm × 30 cm × 40 cm), illuminated by a set of 4 light beams arrays overhead in the horizontal X and Y axes. Mice were placed in the center of the arena and allowed to freely explore for a 30-min period. The track of the mouse was videotaped by a camera fixed on the ceiling of the arena and analyzed with a video-tracking system. The total distance travelled in a 30-min period was calculated and compared in F1$_{GCS}$ as well as its control mice. In the OFT, the area around the center (15 cm × 15 cm) was defined as the central zone. The distances travelled in the center zone of the OFT in a 30-min period were recorded and compared in F1$_{GCS}$ and its control mice.

### Forced swim test (FST).
The FST was performed according to the protocol previously published[67–70]. Briefly, in an FST test, a mouse was held by its tail and gently and slowly placed in a cylindrical tank (18 cm in diameter) filled with 25 cm of water at room temperature (23–27 °C). Once the mouse was in the water, the tail was slowly released to prevent the animal's head from being submerged under the water. On the pre-test day, a mouse was placed in the water for 15 min. On the test day, mice were placed in water for 6 min and their behaviors were recorded by a video camera with high resolution. These records were used later for behavioral scoring. Before returning mice to their home cages, mice were gently wiped using paper towels and placed under a warming lamp until completely dry. 5 mice were tested at the same time with each tank placed in an opaque open field area to prevent mice from seeing each other during the test and potentially altering their behaviors. The mobility in the FST was defined as any movement other than those necessary to balance the body and keep the head above the water. The immobility time served as an index of depression-like behavior.

### Elevated plus maze (EPM).
The EPM was performed according to the protocol previously published[67–70]. Briefly, the apparatus used for the elevated plus maze test comprised two open arms (30 cm × 7 cm × 0.5 cm) across from each other and perpendicular to two closed arms (30 cm × 7 cm × 16 cm) with a center platform (7 cm × 7 cm). A small (0.5 cm) raised lip around the edges of the open arms prevented animals from slipping off, whereas the closed arms have a high (16 cm) wall to enclose the arm. The apparatus was elevated to a height of 100 cm above the floor. A mouse was placed in the center area of the maze with its head directed toward a closed arm, and allowed to freely explore the apparatus for 5 min. Mice were videotaped using a camera fixed above the maze. After each trial, all chambers were cleaned with 75% alcohol and water to prevent a bias based on olfactory cues. Each mouse received one trial in our test battery. The time spent in open and closed arm was scored. An entry in each arm was defined as the two hind legs of the mouse entering the arm. The time spent in the open arms served as an index of anxiety-like behaviors.

### Light–Dark box test (L/D box).
The L/D box was performed according to the protocol previously published[67–70]. Briefly, in the light-dark chambers, a dark (2 lux) and a light chamber (300 lux) were separated to equal sizes by a partition with a small opening at floor level. A mouse was introduced to the dark chamber and habituated for 2 minutes before the trap door was opened. The mouse was allowed to move freely between the two chambers for 5 min with the door open and videotaped using a camera fixed above the maze. After each trial, all chambers were cleaned with 75% alcohol and water to prevent a bias based on olfactory cues. Each mouse received one trial in our test battery. The time spent in the light and dark chamber were scored. The time spent in the light chamber served as the index of anxiety-like behaviors.

## Hippocampal dissection

The dissection procedures of the hippocampus for bulk RNA-Seq, snRNA-Seq, and snATAC-Seq were conducted independently, and all operations were performed at roughly equivalent circadian times. Following deep anesthesia with isoflurane and confirmation of absent pedal reflexes, mice were decapitated. The skull was opened longitudinally along the midline with fine dissection scissors, and the whole brain was carefully extracted and immediately transferred to a Petri dish filled with ice-cold PBS. For all bulk RNA-Seq, we used the entire freshly dissected hippocampus for subsequent experiments. In the paired single-nucleus assays of snRNA-Seq and snATAC-Seq, the isolated hippocampal tissues were snap-frozen in liquid nitrogen for subsequent experiments. Specifically, the left hippocampus from each of six mice was processed for snRNA-Seq library preparation, while the contralateral right hippocampus was used for snATAC-Seq, enabling within-animal comparison across epigenomic and transcriptomic layers. During quality control, one sample was excluded from both datasets due to insufficient library yield or low nuclear integrity, resulting in five high-quality biological replicates per group.

## RNA-Sequence (RNA-Seq) and bioinformatics processing of the data

Total RNA was extracted from the hippocampi of F1$_{GCS}$ and F1$_{CON}$ mice with 3 mice per group at each age at key postnatal developmental stages, including P0 (newborn), P10 (peak of synaptogenesis), and P30 (near maturity), using the RNeasy Mini Kit (QIAGEN, Germany) for bulk RNA-seq analysis, enabling the capture of transcriptional dynamics across development. The RNA quality was assessed using the Bioanalyser 2100 RNA 6000 Nano Kit (Agilent Technologies, USA). mRNA-Seq libraries were generated from total RNA with polyA$^+$ selection of mRNA using the TruSeq RNA Sample Prep Kit v2 (Illumina, USA), and transcriptomes were sequenced using the HiSeq 2000 Sequencing System (Illumina) in paired-end mode. Sequencing adapters and low-quality sequencing reads were

excluded using the Trim Galore program (https://github.com/FelixKrueger/TrimGalore). TopHat2 was used to align the reads to the University of California, Santa Cruz (UCSC) mouse mm10 reference genome. The gene expression matrix was extracted by StringTie. Differential expression analysis was performed by DESeq2 with the thresholds set at $|\log_2FC| > 0.1375$ and adjusted $p$ value < 0.05.

### snRNA-Seq with DNBelab C4 system

To investigate the cell-type-specific molecular foundations of the stable behavioral differences observed in adulthood, we performed snRNA-Seq and snATAC-Seq at P60, a time point when the nervous system has reached maturity and the behavioral phenotypes are well established[71]. Hippocampal samples from F1$_{CON}$ and F1$_{GCS}$ mice ($n = 5$, hippocampi from 6 mice per group, 1 hippocampus was filtered out due to data quality) were collected at this adult stage. Single-nucleus isolation was performed as previously described[72]. In brief, tissues were thawed, minced and transferred to a 2 mL Dounce homogenizer (Sigma-Aldrich, D8938) with 1 mL of homogenization buffer A containing 250 mM sucrose (Sigma-Aldrich, S8501), 10 mg mL$^{-1}$ BSA, 5 mM MgCl2, 0.12 U μL$^{-1}$ RNasin (Promega, N2115) and 1× Complete Protease Inhibitor Cocktail (Roche, 11697498001). Frozen tissues were kept in an ice box and homogenized by 25–50 strokes of the loose pestle (pestle A), after which the mixture was filtered using a 100 μm cell strainer into a 1.5 mL tube. The mixture was then transferred to a clean 1 mL Dounce homogenizer with 750 μL of buffer A containing 1% Igepal (Sigma-Aldrich, CA630), and the tissue was further homogenized by 25 strokes of the tight pestle (pestle B). The mixture was then filtered through a 40 μm strainer into a 1.5 mL tube and centrifuged at 500 g for 5 min at 4 °C to pellet the nuclei. The pellet was resuspended in 1 mL of buffer B containing 320 mM sucrose, 10 mg mL$^{-1}$ BSA, 3 mM CaCl2, 2 mM magnesium acetate, 0.1 mM EDTA (Thermo Fisher Scientific, 15575020), 10 mM Tris-HCl (Invitrogen, AM9856), 1 mM DTT (Invitrogen, 707265 ML), 1× Complete Protease Inhibitor Cocktail and 0.12 U μL$^{-1}$ RNasin. This was followed by centrifugation at 500 $g$ for 5 min at 4 °C to pellet the nuclei. The nuclei were then washed twice with prechilled PBS supplemented with 0.04% BSA and finally resuspended in PBS at a concentration of 1000 nuclei per μL for library preparation.

The DNBelab C Series Single-Cell Library Prep Set (MGI, 1000021082) was utilized as previously described[66]. In brief, the single-nucleus suspensions were converted to barcoded snRNA-Seq libraries through droplet encapsulation, emulsion breakage, mRNA-captured bead collection, reverse transcription, cDNA amplification, and purification. Indexed sequencing libraries were constructed according to the manufacturer's instructions. Library concentrations were quantified using the Qubit ssDNA Assay Kit (Thermo Fisher Scientific, Q10212). Libraries were sequenced using the DIPSEQ T1 sequencer.

### snATAC-Seq with DNBelab C4 system

snATAC-Seq libraries were prepared using the DNBelab C Series Single-Cell ATAC Library Prep Set (MGI, 1000021878)[34]. After Tn5 tagmentation, transposed single-nucleus suspensions were converted to barcoded snATAC-Seq libraries through droplet encapsulation, pre-amplification, emulsion breakage, captured bead collection, DNA amplification, and purification. Indexed libraries were prepared according to the manufacturer's instructions. Concentrations were measured with a Qubit ssDNA Assay Kit (Thermo Fisher Scientific, Q10212). Libraries were sequenced by a DIPSEQ T1 sequencer.

### Computational analysis of snRNA-Seq sequencing data

**Raw data processing.** Raw sequencing reads were filtered and demultiplexed using PISA (version 0.2) (https://github.com/shiquan/PISA). Processed reads were then aligned to the complete mouse reference genome (mm10) using the STAR splicing-aware aligner with default parameters (version 2.7.4a)[73]. For snRNA-Seq, a cell versus gene UMI count matrix was generated with PISA.

**Integration, clustering, and cell type annotation.** Doublet detection and clustering analysis were performed by DoubletFinder (v2.0.3)[74] and Seurat (v4.4.0) R toolkit84 (R version 4.0.2)[75]. NormalizeData, FindVariableGenes, and ScaleData were performed, respectively in each hippocampus sample. The count matrix for nuclei was filtered by the number of unique molecular identifiers (UMIs) > 500, 12,000 > gene > 250, and mitochondria content < 20%. The top 20 dimensions (ranging from dims = 1 to 20) were chosen for the construction of the SNN network. Subsequently, cell clusters were identified through a graph-based clustering approach, specifically the Louvain algorithm (with a resolution of 1). Uniform manifold approximation and projection (UMAP) was employed to visualize the distance between cells in a two-dimensional space. The clustering results were further annotated by highly expressed genes and canonical markers (Supplementary data 2).

**Differential gene expression and functional enrichment analysis.** The FindAllMarkers and FindMarkers functions of Seurat are utilized for differential expression genes (DEGs) analysis between clusters and samples in F1$_{CON}$ and F1$_{GCS}$ with the thresholds set at $|\log_2FC| > 0.1375$ and adjusted $p$ value < 0.05 (Supplementary data 3). The DEGs obtained from each comparison were input into the Metascape online tool[76] for functional enrichment analysis ($p < 0.01$, count > 3, and enrichment factor > 1.5). The results were visualized by Bar plot[77].

**Cell-cell interaction analysis.** CellChat (v2.1.2)[37] was used to detect ligand-receptor interactions in integrated snRNA-Seq data following standard procedures. The CellChat database (CellChatDB) was retrieved from http://www.cellchat.org/cellchatdb/. Analyses were performed on log-transformed normalized gene expression matrices using default parameters. For each group, data were input into the CellChat pipeline. Intercellular communication probabilities and networks were inferred using computeCommunProb with population.size = TRUE to account for differences in cell population sizes. To identify differential cell-cell communication events between F1$_{CON}$ and F1$_{GCS}$ groups, we merged datasets using mergeCellChat, compared overall communication strength using compareInteractions, and visualized significant differences in interaction probabilities using netVisual_diffInteraction. A heatmap depicting changes in signal strength was generated using netVisual_heatmap.

**Association analysis of single nuclei with diseases and complex features.** Single-cell disease relevance score (scDRS)[48] is a method that links sc/snRNA-Seq to polygenic disease risk at single-cell resolution, irrespective of annotated cell type. The diseases included in this study are attention deficit hyperactivity disorder (ADHD), Alzheimer's disease (AD), bipolar disorder (BIP), drinks per week, general risk tolerance (GRT), insomnia, intelligence (Intel), major depressive disorder (MDD), reaction time, schizophrenia (SCZ), sleep duration, subject well-being (SWB), verbal numerical reasoning (VNR), and anxiety. To comprehensively evaluate the potential long-term neuroprotective effects of choline supplementation, we performed downstream scDRS analyses based on the statistical significance ($p$-value) of disease associations at the single-cell level. Cell-type-level analyses were also conducted to assess associations between predefined hippocampal cell subpopulations and each disease. Additionally, the heterogeneity in disease relevance among individual cells within each cell type was evaluated.

### Computational analysis of snATAC-Seq sequencing data

**Data processing and analysis.** FASTQ files generated from sequencing were aligned to reference genomes (mm 10) using chromap (v0.2.4)[78]. Sequencing barcodes were assigned to cell barcodes, and fragment files were generated using d2c (v1.4.4, available at https://github.com/STOmics/d2c). Downstream analysis of snATAC-Seq data was conducted using ArchR (v1.0.2)[79]. Specifically, nuclei with transcription start site (TSS) enrichment scores < 4 and fragment counts < 1,000 were filtered out. Doublets were also

removed using the filterDoublets function with filterRatio = 1.5. Next, dimensionality reduction of the entire genome was performed using the addIterativeLSI function based on latent semantic indexing, with parameters set to iterations = 2, resolution = 1.5, and dimsToUse = 1–50. We integrated the snATAC-Seq data of each cell type with the corresponding snRNA-Seq data using the addGeneIntegrationMatrix function in ArchR. By anchoring the snATAC-Seq and snRNA-Seq datasets, we annotated the cell types for the snATAC-Seq data.

**Gene activity score**. We utilized the default model in ArchR to calculate gene activity scores based on the accessibility of gene bodies, promoters, and distal regulatory elements, and associated these scores with gene expression using default parameters.

**Motif enrichment, differential gene accessibility, and functional enrichment analysis**. Before motif enrichment analysis, a reproducible peak set was generated in ArchR using the addReproduciblePeakSet function, stratified by cell types. Motif annotations were added using the addMotifAnnotations function with motifSet = "cisbp". Peaks for each cluster were called using macs2[80] via the addRepresentativePeakSet function. Differential peaks for each cluster were identified using the getMarkerFeatures function with thresholds of $|\log_2 FC| > 0.1375$ and FDR < 0.05. Genes annotated to these differential peaks were defined as differentially accessible genes (DAGs). The DAGs obtained from each comparison were input into the Metascape online tool[76] for functional enrichment analysis ($p < 0.01$, count > 3, and enrichment factor > 1.5). The results were visualized by Bar plot[77].

**RNA extraction and RT–qPCR analysis**. To extract the total RNAs, the hippocampi were collected and lysed with 500 µL TRIzol reagent, followed by the addition of 100 µL chloroform. After centrifugation (12,000 rpm for 15 min at 4 °C), the total RNAs in the supernatant were precipitated with isopropanol. The cDNAs were reverse transcribed with the HiScript III 1st Strand cDNA Synthesis Kit (+gDNA wiper) (Vazyme, R312-01). qRT-PCR was performed using ChamQ Universal SYBR qPCR master mix (Vazyme, Q711-02). All qRT-PCR results were analyzed by the Mann-Whitney U test. Gene expression levels were normalized to *β-Actin*. The primer sequences used in this study were provided in Supplementary data 5.

**Western blotting**. To examine the levels of DCX, proteins were extracted from the hippocampus using the Animal Tissue/Cell Total Protein Extraction Kit (BC3790, Solarbio, China) according to the manufacturer's instructions. Protein samples were loaded onto 4–20% FuturePAGE™ precast gels (ET12420Gel, ACE, China) for 1 h and transferred to a PVDF membrane at 300 mA for 1 h. The membranes were blocked with 5% non-fat milk for 1 h at room temperature and then incubated overnight at 4 °C with the following primary antibodies: β-Tubulin (A12289, 1:10,000, Abclonal, China), Doublecortin (YM8640, 1:6000, Immunoway, USA), Membranes were washed and then incubated with a horseradish peroxidase (HRP)-conjugated secondary antibody (7074S, 1:5000, Cell Signaling Technology, USA) for 1 hour at room temperature.

**Statistics and reproducibility**. For behavioral tests, each group consisted of 14 biological replicates. For bulk transcriptome sequencing, each group included 3 biological replicates. For single-nucleus omics analyses, each group comprised 5 biological replicates. For qRT-PCR, each group was represented by 3 biological replicates across three developmental stages. For western blotting, each group contained 3–6 replicates at P0 and P60. Results are presented as means ± standard error of the means. Statistical tests were performed using either an unpaired two-tailed Student's *t* test, the Mann-Whitney U test, or the Wilcoxon test after the analysis of homogeneity of variance and normal distribution. For omics analyses, *p*-values were further adjusted using the Benjamini-Hochberg method to correct for multiple testing errors. All statistical analyses were performed using GraphPad Prism (La Jolla) and R version 4.3.

## Data availability

All raw data have been deposited in the Genome Sequence Archive (GSA; accession number PRJCA043148) at the BIG Data Center, Beijing Institute of Genomics, Chinese Academy of Sciences. The processed snRNA-seq data are accessible via OMIX at the China National Center for Bioinformation/ Beijing Institute of Genomics, Chinese Academy of Sciences (accession no. OMIX011041). Numerical source data for all graphs and charts are provided in the supplementary information and have been deposited at figshare (https://doi.org/10.6084/m9.figshare.31441492).

## Code availability

All custom code used for the analyses presented in this paper is included in the Figshare dataset deposited alongside the source data. It is freely available under the https://doi.org/10.6084/m9.figshare.31441492.

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

## Acknowledgements

This work was supported by the National Key R&D Program of China (No. 2025YFA1309200), National Natural Science Foundation of China (Nos. 32270657, 32470630, and 32570726), the Natural Science Foundation of Beijing Municipality (No. 5242018), and State Key Laboratory of Animal Biodiversity Conservation and Integrated Pest Management (No. SKLA2508). The authors have no relevant financial or non-financial interests to disclose.

## Author contributions

Conceptualization: Y.W. and Z.S., Methodology: X.S., Y.L., M.W., X.Z., M.L., Q.L., W.X., Z.S., Y.W., Investigation: X.S., Y. L., M.W., X. Z., Q.L., W.X., Z.S., Y.W., Visualization: X.S. and Y.W., Supervision: Y.W. and Z.S. Writing original draft: X.S. and Y.W., Writing editing and review: X.S., Y.L., M.W., X.Z., M.L., Q.L., W.X., Z.S., Y.W.

## Competing interests

The authors declare no competing interests.
