## [Transparent Peer Review file · Communications Biology]

Gestational choline supplementation regulates hippocampal granule neuron development and emotion-like behavior

Corresponding Author: Professor Yan Wang

Version 0:

Reviewer comments:

Reviewer #1

(Remarks to the Author)

In this study, Shi et al. assessed the impact of choline supplementation during gestation on hippocampus development in the mouse. To achieve their aim, the authors used a combination of omics approaches (bulk mRNA-sequencing, single-nucleus RNA sequencing and single-nucleus ATAC-sequencing) and behavioural tests. Their main claim is that choline supplementation during gestation increases the proportion of immature granule neurons in the hippocampus (while decreasing that of mature granule neurons), with significant impact on the transcriptome and chromatin accessibility of each cell type, as well as on the intercellular communication. They specifically highlight the reduced expression of *Mif* in immature granule neurons and link that with the glucocorticoid signalling pathway, providing a potential connection with the anxiolytic responses observed in the offspring of mothers exposed to choline supplementation during pregnancy. The study is overall interesting and original in its approach, and the methodology used is powerful. However, I find it difficult to follow the data in the way it is presented. In addition, I think some of the main claims are based on changes that are very small in magnitude and without consideration of potential false positives. In addition, other main claims are solely based on *in silico* predictions. Confirmation by alternative approaches would increase the confidence and improve the impact of the paper.

I have several points that I think the authors should consider:

1. The authors seem to have performed all the experiments in male offspring. This should be clearer (maybe even in the abstract) and a reason for excluding female offspring should be given. There is growing evidence for sex-specific intergenerational effects, and this limitation should be discussed. The exclusion of female offspring may have additional ramifications for the paper. For example, when the authors are attempting to integrate their findings with those from GWAS studies, how confident are they that sex does not play a role?
2. The definition used for DEGs seems very relaxed: a $\text{Log}_2\text{FC} < 0.1375$ and $p < 0.05$ (mentioned in the methods) means that the authors are confident that they can measure accurately changes that are just greater than 10% and without applying any FDR. I do not share this level of confidence. I think the authors should apply more stringent cutoffs and FDR for all the omics analyses. How many of the effects currently highlighted would survive is unclear and the main message of the paper may suffer significant changes.
3. How do the authors define a "differentially accessible gene" (DAG)? There is no "definition" given in the methods.
4. In Figure S1E the authors present the overlap between DEGs and DAGs, however is that overlap significantly higher than one would expect by random chance?
5. In the single-nucleus data, the claim that there is an increase in the proportion of immature granule neurons in the hippocampus of offspring exposed to choline supplementation needs to be validated through additional means. I'm not convinced that a single marker gene for mature granule neurons (*Snhg11*) and two marker genes for immature granule neurons (*Dsp* and *Gda*) is enough to classify the single nuclei with enough confidence, given the inherent limitations of single-nucleus sequencing techniques). Although both sn-RNA-seq and sn-ATAC-seq suggest an increase in the proportion of immature granule neurons in offspring exposed to choline supplementation, the proportions shown in Figure 3A are quite different. For example, mature granule neurons are in the range of 30-40% according to snRNA-seq and around 15% according to sn_ATAC-seq. Which of these values do I believe? Confirming the changes in the proportion of cells by histological means would be very beneficial.
6. Using CellChat, the authors predict that pre-natal choline supplementation enhances intercellular communication in immature granule neurons. This is another section of the paper that I think would benefit greatly from validation through complementary approaches. For example, the authors claim that immature granule neurons have an enhanced communication via the glutamate pathway. I find the illustration shown in Figure 4D unconvincing (there may be some subtle differences, but can they be compared through statistical means?). Have the authors measured any changes in the

glutamate levels in the hippocampi of the offspring exposed to prenatal choline supplementation? Mif binds to several receptors, such as Cd74, Cxcr2 and Cxcr4. Are these differentially expressed?

7. The authors mention in the next section a link between Mif and the glucocorticoid response pathways. However, in the heatmap presented in Figure 5A, the only effect that is clearly visible is the impact of postnatal age (P0 versus P10 versus P60). I cannot see any clear difference between the two maternal diets for any of the sets of genes depicted. The volcano plots shown in panels 5B-5D illustrate again one of the points made above that most of the “DEGs” have very small fold changes and that there is no FDR applied to correct the data.

8. The changes observed at 2 months of age by sn-ATAC-seq, when the mice are no longer exposed to the choline supplementation suggest possible epigenetic memory. The authors have not performed ATAC-seq at earlier time points (as they do for transcriptome, using bulk-mRNA-seq) to see if regions of open chromatin are established early and then maintained. Have the authors tried to look at genes that are DEGs early and are then maintained as “differentially expressed”? I don't see any clear evidence for that in data presented in Figure 5.

Additional points:

a. The authors should settle for a more uniform terminology when referring to the impact on behavioural changes. In the paper, they are using: “despair-related response” (can one even refer to “despair” in a mouse?), “anxiety-related behaviour”, “mood-related traits”, “depression-like responses”. I think this can be simplified.

b. How confident are the authors that the choline supplementation reaches the brain of the developing fetuses? Have they measured it?

c. The authors give accession numbers for two repositories where they submitted their sequencing data. However, I tried both and neither is accessible. The role of a reviewer is to look at all the aspects of a paper and assessing raw sequencing data is part of it.

Reviewer #2

(Remarks to the Author)

The study by Shi et al. investigates the effects of gestational choline supplementation on multiple complex parameters in the hippocampus, using single-nucleus RNA sequencing, single-nucleus assay for transposase-accessible chromatin sequencing and anxiety-like behavior. Some analyses were conducted at different developmental stages using bulk RNA sequencing. The authors also extrapolated certain data to explore intercellular communication. Additionally, they performed a correlation analysis to derive a single-cell “disease relevance score”. The work is relevant, addressing the impact of maternal nutrition on offspring development. However, several methodological details and the rationale behind some experimental steps remain unclear. A notable limitation is the largely descriptive nature of the study, lacking definitive functional data. Except for some behavioral tests, many parameters related to cell communication appear speculative, relying on indirect measures such as “relative or differential interaction strength”.

Specific comments

1. What is the rationale behind the chosen dosage of the choline-supplemented diet?

2. How did the authors determine the number of animals used in each analysis and at each developmental stage?

3. How many dams (litters) were included in the study, and what were the sizes of individual litters? Could choline supplementation have differential effects on offspring depending on litter size? What was the sex ratio within each litter? Given that choline supplementation is known to influence neuronal number and differentiation during prenatal development, could this affect postnatal changes observed in the offspring?

4. Why was only the right hippocampus dissected for analysis? What were the coordinates and size of the sampled tissue? Could tissue size or the number of neurons and glia impact the analytical results? The tissue sampling method is critical for determining sampling bias, as some cells may not have been captured or analyzed, and different hippocampal subregions may exhibit significant regional variability.

5. Considering the low sample size at each developmental stage, what do the authors consider the biological variable for the analysis...the litter or the individual animal? Clarification on this point would be valuable.

6. Although the cell communication analysis is interesting, it is limited to the hippocampus and does not provide functional insight into behavior, which is influenced by other brain regions.

7. The correlation of gene expression changes in the used animal model with databases related to Alzheimer's disease, bipolar disorder, insomnia, major depressive disorder, and schizophrenia may be of potential interest, but in this study it appears only partially justified and insufficiently explained. What was the rationale for including these comparisons? Were they driven simply by data availability, or was there a specific hypothesis related to nutritional intervention?

Reviewer #3

(Remarks to the Author)

In this manuscript, Wang and colleagues examine the impact of gestational choline supplementation (GCS) on the hippocampal development using a mouse model. The authors examined the transcriptional alterations and chromatin accessibility modulated by GCS at the single-nucleus level using multi-omics approaches, such as single-nucleus RNA sequencing (snRNA-seq) and single-nucleus assay for transposase-accessible chromatin sequencing (snATAC-seq). The author found that the transcriptome, intercellular communication, and chromatin accessibility of immature granule neurons were altered in the hippocampus of male offspring with GCS. Using bulk transcriptome analysis of the hippocampus, the authors also demonstrated that GCS modulates neuronal development and biological events during critical developmental periods. Furthermore, single-cell disease relevance analysis (scDRS) revealed an association between the gene expression profiles of immature granule neurons in the hippocampus of F1 GCS mice and various diseases, including major depressive disorder and anxiety. The reviewer commended the study's robust experimental design and rigorous

bioinformatic analyses, including integrative approaches. This manuscript represents a significant contribution to the field of developmental neuroscience, setting high standards for experimentation and analysis. It also emphasizes key biological and analytical considerations for such studies while introducing gene expression variability at the single-cell level as a potential explanation for how nutritional supplementation affects central nervous system development. However, the manuscript remains incomplete in some areas that could be improved.

Major concerns:

1) Experimental design:

a) In this study, pregnant female mice were fed a choline-supplemented diet from gestational day 9 until parturition.

However, I did not find a description of the reason for the feeding period. The authors need to explain why they chose this feeding period.

b) The authors collected the hippocampus from F1 mice at P0, P10, and P30 for bulk RNA-seq, and at P60 for snRNA-seq and snATAC-seq. As mentioned above, there is no explanation for why the samples were collected at these time points. The authors should explain why they chose these time points to collect the hippocampus.

2) Lacking the experimental evidence of cell proportion change:

The authors demonstrated an increased proportion of immature granule neurons and a decreased proportion of mature granule neurons in the hippocampus of F1 GCS mice at the gene expression level. Has a previous study investigated the proportion change modulated by GCS more direct method such as immunofluorescence staining of marker proteins? If not, the authors should consider performing the experiments showing the proportion change of granule neurons (mature to immature state) modulated by GCS at P60 to reinforce the conclusion.

3) Consider the sex differences:

In this study, the authors examined the effects of GCS on the hippocampal development in male F1 mice. Why were only male offspring analyzed? Many models nowadays exhibit dysmorphic responses, so it would greatly enrich the work if female offspring were also studied. If not, the authors should explain their reasoning in the manuscript.

Minor concerns:

1) In the title of the manuscript, I would like you to consider the inclusion of "hippocampus (or hippocampal)".

2) It is not clear if the hippocampus was dissected after or before freezing the brain. Detailed method for dissection of the hippocampus should be clarified to ensure reproducibility.

3) In Fig. S3A, asterisk is disappeared.

4) Page 24, Line 5, "The time in open and closed arm were scored" should be revised as "The time spent in light and dark chamber were scored".

Version 1:

Reviewer comments:

Reviewer #1

(Remarks to the Author)

In this revised manuscript, Xiaohui Shi et al. have performed additional experiments and analyses to address the points raised by me and my co-reviewers during the first round of review. The authors have also undertaken substantial editing of the text. Overall, these changes have resulted in a significantly improved and clearer manuscript. Nevertheless, several aspects still require further attention. Below, I refer strictly to the points I raised in the first round of review and leave the assessment of my colleagues' comments to the other reviewers.

1. FDR correction across omics datasets

The consistent application of FDR correction across all omics datasets represents an improvement (even though the fold-change threshold remains relatively low). However, the authors need to ensure that this change is reflected consistently across all relevant figure panels and corresponding figure legends. For example, in Figure 2 (F–H), Figure 3 (F–G), and Figure S4 (A–C), the y-axis is still labeled as “ $-\log_{10}$ (p value)” rather than “ $-\log_{10}$ (FDR-corrected p value)” or “ $-\log_{10}$ (q value)”.

2. Statistical analysis of newly added experimental data

For the newly added experimental data presented in several figures (Figure 3B–E, Figure S2E, Figure S3, Figure 4F, Figure 4H, and Figure 5B), the authors should apply non-parametric statistical tests (e.g., Mann–Whitney tests) when $n = 3$ per group. The use of t-tests assumes normality, which cannot be reliably assessed with such small sample sizes. That said, I anticipate that the overall interpretation of the results will remain largely unchanged.

3. Interpretation of persistent differentially expressed genes

With regard to the two genes—Lpar1 and Ttr—that show differential expression emerging at P10 and persisting through P60, it would be helpful to clarify whether these genes fall into the category of what the authors term “differentially accessible genes (DAGs).” If so, altered chromatin accessibility could represent a plausible mechanism of epigenetic memory linking early exposure to later outcomes. It would therefore be valuable to reference these two genes explicitly in the Discussion, particularly if they have known biological relevance to the observed phenotypes.

Reviewer #2

(Remarks to the Author)

I have no further questions. The manuscript has been satisfactorily improved.

Reviewer #3

(Remarks to the Author)

The authors were responsive to the reviewer comments. I endorse publication of this paper, which is an important contribution to the field.

Response to the Reviewers:

Reviewer #1 (Remarks to the Author):

In this study, Shi et al. assessed the impact of choline supplementation during gestation on hippocampus development in the mouse. To achieve their aim, the authors used a combination of omics approaches (bulk mRNA-sequencing, single-nucleus RNA sequencing and single-nucleus ATAC-sequencing) and behavioural tests. Their main claim is that choline supplementation during gestation increases the proportion of immature granule neurons in the hippocampus (while decreasing that of mature granule neurons), with significant impact on the transcriptome and chromatin accessibility of each cell type, as well as on the intercellular communication. They specifically highlight the reduced expression of Mif in immature granule neurons and link that with the glucocorticoid signalling pathway, providing a potential connection with the anxiolytic responses observed in the offspring of mothers exposed to choline supplementation during pregnancy. The study is overall interesting and original in its approach, and the methodology used is powerful. However, I find it difficult to follow the data in the way it is presented. In addition, I think some of the main claims are based on changes that are very small in magnitude and without consideration of potential false positives. In addition, other main claims are solely based on in silico predictions. Confirmation by alternative approaches would increase the confidence and improve the impact of the paper.

I have several points that I think the authors should consider:

1. *The authors seem to have performed all the experiments in male offspring. This should be clearer (maybe even in the abstract) and a reason for excluding female offspring should be given. There is growing evidence for sex-specific intergenerational effects, and this limitation should be discussed. The exclusion of female offspring may have additional ramifications for the paper. For example, when the authors are attempting to integrate their findings with those from GWAS studies, how confident are they that sex does not play a role?*

Response: We sincerely appreciate these insightful comments regarding the potential implications of excluding female offspring. We agree that our study design, while methodologically justified to minimize potential confounding effects associated with hormonal fluctuations in females (PMID: 39002829, 15642623, 31253786), limits the generalizability of our findings to male offspring only. Accordingly, we have clearly specified the sex used in the revised Abstract (Page 2, Line 5), Methods (Page 28, Line 17), and Results sections. This limitation is particularly relevant given the growing evidence for sex-specific intergenerational effects and the integration of our study findings with GWAS research. We thus have added a specific discussion regarding neural mechanisms we report may not fully extend to female populations (Page 27, Lines 20–25).

2. *The definition used for DEGs seems very relaxed: a $\text{Log}_2\text{FC} < 0.1375$ and $p < 0.05$ (mentioned in the methods) means that the authors are confident that they can measure accurately changes that are just greater than 10% and without applying any FDR. I do not share this level of confidence. I think the authors should apply more stringent cutoffs and FDR for all the omics analyses. How many of the effects currently highlighted would survive is unclear and the main message of the paper may suffer significant changes.*

Response: We appreciate the reviewer’s valuable comments regarding the robustness of the currently identified effects. In response, we have applied more stringent statistical thresholds, including false discovery rate (FDR) correction, to all multi-omics datasets, as detailed in the Methods section (Page 31, Lines 11–12; Page 33, Lines 9–10 and Page 35, Line 5). Specifically, for both RNA-Seq and snRNA-Seq data, the criteria for identifying differentially expressed genes (DEGs) have been updated to $|\log_2(\text{fold change})| > 0.1375$ and adjusted p value < 0.05 . Our re-analysis indicates that while the number of significant effects has been reduced the core findings of the study remain substantively unchanged. All relevant figures and tables—including Fig. S1C–E, Fig. 2F–H, Fig. 3G–H, Fig. S4, Fig. 5B–D, and Table S3—have been updated accordingly in the revised manuscript.

3. *How do the authors define a “differentially accessible gene” (DAG)? There is no “definition” given in the methods.*

Response: We thank the reviewer for this thoughtful suggestion. A clear definition of “differentially accessible gene (DAG)” have been provided in the Materials and Methods section (Page 35, Lines 5–6) in the revised manuscript, as follows: “Differential peaks for each cluster were identified using the getMarkerFeatures function with thresholds of $|\log_2\text{FC}| > 0.1375$ and $\text{FDR} < 0.05$. Genes annotated to these differential peaks were defined as differentially accessible genes (DAGs). The DAGs obtained from each comparison were submitted to the Metascape online tool for functional enrichment analysis ($p < 0.01$, count > 3 , and enrichment factor > 1.5)”.

4. *In Figure S1E the authors present the overlap between DEGs and DAGs, however is that overlap significantly higher than one would expect by random chance?*

Response: We appreciate the reviewer’s thoughtful suggestion. To evaluate whether the overlap between differentially expressed genes (DEGs) and differentially accessible genes (DAGs) in Figure S1E occurs beyond random expectation, we conducted hypergeometric and Fisher’s exact tests for both up- and down-regulated features across all cell types and observed no statistically significant enrichment in any cell types (all adjusted $p > 0.05$). These results suggest the involvement of intermediate regulators, such as transcription factors or post-translational modifications, that mediate chromatin accessibility and gene expression changes induced by gestational choline supplementation. This analysis and related interpretation have been added in the relevant section of the revised manuscript (Page 6, Lines 27–29 and Page 7, Lines 1–3).

5. *In the single-nucleus data, the claim that there is an increase in the proportion of immature granule neurons in the hippocampus of offspring exposed to choline supplementation needs to be validated through additional means. I’m not convinced that a single marker gene for mature granule neurons (Snhg11) and two marker genes for immature granule neurons (Dsp and Gda) is enough to classify the single nuclei with enough confidence, given the inherent limitations of single-nucleus sequencing techniques). Although both sn-RNA-seq and sn-ATAC-seq suggest an increase in the proportion of immature granule neurons in offspring exposed to choline supplementation, the proportions shown in Figure 3A are quite different. For example, mature granule neurons are in the range of 30-40% according to snRNA-seq and around 15% according to sn_ATAC-seq. Which of these values do I believe? Confirming the changes in the proportion of cells by histological means would be very beneficial.*

Response: We thank the reviewer for this critical suggestion regarding the validation of cell type proportions in our single-nucleus data. We acknowledge the discrepancy in the proportion estimates between snRNA-Seq and snATAC-Seq, which likely stems from differences in assay sensitivities, cell state detection, and data normalization methods. To address this concern, we have performed additional validation using Western blotting and quantitative real-time PCR (qRT-PCR) with established markers for immature and mature granule neurons in the revised manuscript. These results confirm the increase in the proportion of immature granule neurons following gestational choline supplementation at both P0 and in adulthood, and the results have been incorporated into the revised manuscript to strengthen our conclusions (Fig. 3B–E; Page 10, Lines 20–25).

6. *Using CellChat, the authors predict that pre-natal choline supplementation enhances intercellular communication in immature granule neurons. This is another section of the paper that I think would benefit greatly from validation through complementary approaches. For example, the authors claim that immature granule neurons have an enhanced communication via the glutamate pathway. I find the illustration shown in Figure 4D unconvincing (there may be some subtle differences, but can they be compared through statistical means?). Have the authors measured any changes in the glutamate levels in the hippocampi of the offspring exposed to prenatal choline supplementation? Mif binds to several receptors, such as Cd74, Cxcr2 and Cxcr4. Are these differentially expressed?*

Response: We thank the reviewer for the constructive comments. To address the statistical concern in intercellular communication, we have performed a quantitative comparison of communication probabilities using unpaired *t* test on the interaction strength of immature granule neurons as both source and target cells. The results indicated that cell-cell communication was significantly enhanced in F1_{GCS} mice compared to controls (Fig. 4C; Page 13, Line 24).

Regarding glutamate level measurement, our snRNA-Seq data, supported by qRT-PCR validation, showed significant upregulation of a set of genes, including *Gria1*, *Gria2*, and *Grik5* in the glutamate pathway in the hippocampus of F1_{GCS} mice at both postnatal day 0 (P0) and adulthood (Fig. S2E, Fig. S3, Fig. 4F; Page 14, Lines 9–14).

For the MIF signaling pathway, our snRNA-seq data and RT-PCR validation showed down-regulated expression of *Mif* and its receptors, such as *Ackr3*, *Cd74*, *Cxcr2* and *Cxcr4* in the hippocampus of F1_{GCS} mice at postnatal day 0 (Fig. 4H, Fig.S3D; Page 14, Lines 18–23). The results have been incorporated into the revised manuscript to strengthen our conclusions.

7. *The authors mention in the next section a link between Mif and the glucocorticoid response pathways. However, in the heatmap presented in Figure 5A, the only effect that is clearly visible is the impact of postnatal age (P0 versus P10 versus P60). I cannot see any clear difference between the two maternal diets for any of the sets of genes depicted. The volcano plots shown in panels 5B-5D illustrate again one of the points made above that most of the “DEGs” have very small fold changes and that there is no FDR applied to correct the data.*

Response: We thank the reviewer for the insightful observation. As correctly noted, Fig. 5A illustrates the stage-dependent expression patterns of genes involved in key pathways across postnatal hippocampus development. To further characterize stage-specific transcriptional regulation, we have now included functional enrichment analysis of up- and down-regulated differentially expressed genes at each postnatal time point (Fig. S4A-C; Page 19, Lines 3–11). These

results confirm that hippocampal gene expression programs shift dynamically across developmental stages. While the maternal dietary effect is subtler than the developmental changes, we did identify significant diet-associated differences in the glucocorticoid pathway genes. Regarding the statistical thresholding, we have now applied FDR correction (adjusted $p < 0.05$) to all differential expression analyses and updated the volcano plots accordingly in the revised version (Fig. 5B-C). The results have been incorporated into the revised manuscript.

8. *The changes observed at 2 months of age by sn-ATAC-seq, when the mice are no longer exposed to the choline supplementation suggest possible epigenetic memory. The authors have not performed ATAC-seq at earlier time points (as they do for transcriptome, using bulk-mRNA-seq) to see if regions of open chromatin are established early and then maintained. Have the authors tried to look at genes that are DEGs early and are then maintained as “differentially expressed”? I don’t see any clear evidence for that in data presented in Figure 5.*

Response: We thank the reviewer for raising the important question regarding the potential epigenetic mechanism that may underlie long-term regulation of prenatal nutritional interventions in the hippocampus. The observed chromatin accessibility changes at 2 months of age, after choline supplementation has ceased, do suggest a potential epigenetic mechanism. In response to the reviewer’s question, we have conducted additional analyses to identify genes that are differentially expressed at early stages (e.g., P0) and remain so in adulthood. We found two genes—*Lpar1* and *Ttr*—exhibited differential expression that emerged at P10 and persisted through P60 (Fig. S4E–G; Page 19, Lines 11–14). Additionally, we validated that a set of genes in the glutamate- and MIF-related pathways showed consistent alterations in the hippocampus of F1_{GCS} mice at P0 and P60 (Fig. S2E, Fig. 4F, Fig. 4H; Page 14, Lines 9–23). These results suggest that prenatal nutritional interventions may exert stage-specific regulation in brain function and behavior. The results and discussion have been incorporated into the revised manuscript.

Additional points:

a. *The authors should settle for a more uniform terminology when referring to the impact on behavioural changes. In the paper, they are using: “despair-related response” (can one even refer to “despair” in a mouse?), “anxiety-related behaviour”, “mood-related traits”, “depression-like responses”. I think this can be simplified.*

Response: We appreciate the reviewer’s comment. All relevant terms have been uniformly revised to “anxiety or depression-like behavior” in the revised manuscript (Page 2, Line 5; Page 3, Line 20 and 23; Page 4, Line 5, 11, 17, and 23; Page 26, Line 29; Page 27, Line 5 and 16; Page 28, Line 14).

b. *How confident are the authors that the choline supplementation reaches the brain of the developing fetuses? Have they measured it?*

Response: We thank the reviewer for this valuable comment. Our snRNA-Seq data, supported by qRT-PCR validation, showed significant upregulation of a set of genes in the choline-related pathway, including *Chrna3*, *Creb3l2*, *Itp3*, and *Pla2g4b* in the hippocampus of F1_{GCS} mice at P0 (Fig. 5B; Page 19, Lines 17–19), suggesting an effect of choline supplementation. The results have been incorporated into the revised manuscript.

c. The authors give accession numbers for two repositories where they submitted their sequencing data. However, I tried both and neither is accessible. The role of a reviewer is to look at all the aspects of a paper and assessing raw sequencing data is part of it.

Response: We apologize for the inconvenience, and the shared links (<https://ngdc.cncb.ac.cn/gsa/s/yV7I3H0p> and <https://ngdc.cncb.ac.cn/omix/preview/0vWjDBaH>) have been added to the Data Availability section in the revised manuscript (Page 36, Lines 6–11).

Reviewer #2 (Remarks to the Author):

The study by Shi et al. investigates the effects of gestational choline supplementation on multiple complex parameters in the hippocampus, using single-nucleus RNA sequencing, single-nucleus assay for transposase-accessible chromatin sequencing and anxiety-like behavior. Some analyses were conducted at different developmental stages using bulk RNA sequencing. The authors also extrapolated certain data to explore intercellular communication. Additionally, they performed a correlation analysis to derive a single-cell “disease relevance score”. The work is relevant, addressing the impact of maternal nutrition on offspring development. However, several methodological details and the rationale behind some experimental steps remain unclear. A notable limitation is the largely descriptive nature of the study, lacking definitive functional data. Except for some behavioral tests, many parameters related to cell communication appear speculative, relying on indirect measures such as “relative or differential interaction strength”.

Specific comments:

1. *What is the rationale behind the chosen dosage of the choline-supplemented diet?*

Response: We thank the reviewer for their insightful question regarding the rationale for the choline dosage selection. The dosage of choline administered to the pregnant mice in the CON and GCS groups was determined based on a review of existing literature on choline supplementation in various mouse models (PMID:39046330 and PMID:34690739), which have been supplemented in the “Mice and Gestational Choline Supplementation (GCS) Program” section of Materials and Methods in the revised manuscript (Page 28, Lines 7–11).

2. *How did the authors determine the number of animals used in each analysis and at each developmental stage?*

Response: We thank the reviewer for raising this important question regarding sample size determination in our experimental design. The sample sizes for each analysis and developmental stage were determined based on the core principle of ensuring adequate biological replication, while striving to maximize statistical power within the constraints of research resources and ethical considerations for animal use. Specific justifications are as follows:

Behavioral analysis: To prevent potential litter effects, we strictly limited testing to 1–2 male offspring per dam. To maintain statistical power with this design, the total number of litters was increased, yielding a final sample size of $n = 14$ mice per group. This information has been added to the “Mice and Gestational Choline Supplementation (GCS) Program” section in the revised manuscript (Page 28, Lines 14–21). The calculation of Cohen’s d for behavioral tests demonstrated

that 14 mice per group provided sufficient statistical power (OFT: 0.895; EPM: 2.258; L/D box: 1.243; FST: 1.916) to detect meaningful intergroup differences (Fig. 1D-G; Page 5, Lines 7–9).

Bulk RNA-seq at developmental stages: To balance the substantial cost of multi-time-point sequencing with the ability to detect significant transcriptional changes, we set a sample size of $n = 3$ per group for each time point (P0, P10, P30). This meets the minimum requirement for biological replication in transcriptomic studies and allows for robust differential expression analysis using tools such as DESeq2 (Page 31, Lines 1–4).

Single-nucleus sequencing: We adopted a strategy that balances data depth with biological context. The five hippocampal samples per group used for nuclei isolation were derived from the left and right hemispheres of six mice (one hippocampus was excluded due to quality control). This approach enhances cellular diversity in the captured population, helps average out individual variability, and improves the robustness of the single-cell data (Page 30, Lines 21–27).

In summary, our sample size choices were not arbitrary but were carefully considered to ensure scientific validity and reliability, prioritizing biological replication and statistical power while taking into account practical and ethical constraints.

3. How many dams (litters) were included in the study, and what were the sizes of individual litters? Could choline supplementation have differential effects on offspring depending on litter size? What was the sex ratio within each litter? Given that choline supplementation is known to influence neuronal number and differentiation during prenatal development, could this affect postnatal changes observed in the offspring?

Response: We appreciate the reviewer’s insightful comments regarding the experimental design and sample considerations. In response, we have clarified our methodology in the revised manuscript (Page 28, Lines 14–22). To ensure robust and independent biological replication, this study employed a split-experiment design with independent sampling across experimental modules. Male offspring for behavioral testing, omics profiling, and molecular validation were sourced from separate breeding cohorts to eliminate litter effects. Specifically, behavioral analysis ($n = 14$) involved 7–8 litters per group, single-nucleus sequencing ($n = 5$) utilized 2–3 litters per group, bulk RNA-seq ($n = 3$) required 2–3 litters per group, and molecular assays (qRT-PCR/WB; $n = 3–6$) were based on 3–4 litters per group, totaling approximately 14–18 independent litters per group. The sample size for behavioral tests was determined a priori based on power considerations, and calculation of Cohen’s d confirmed large effect sizes (OFT: 0.895; EPM: 2.258; L/D box: 1.243; FST: 1.916), indicating that 14 mice per group provided sufficient statistical power to detect meaningful inter-group differences (Fig. 1D-G; Page 5, Lines 7–9). All dams were maintained under identical gestational conditions. Litter size varied between 7–10 pups, with a sex ratio close to 1:1, consistent with the C57BL/6J strain’s typical reproductive pattern. From each litter, only 1–2 male pups were randomly selected for downstream analyses to preclude confounding by the female estrous cycle (PMID: 15642623, 39002829, 31253786). This multi-litter random sampling strategy averaged inter-litter variability within groups, confirming that neither litter size nor sex ratio substantially affected the observed outcomes of choline supplementation.

The supplementation window (E9 to birth) coincides with the critical phase of hippocampal granule neuron proliferation and differentiation (PMID: 28891434, 25032496, 36504449). Our findings indicate

that prenatal choline supplementation induces lasting changes in postnatal hippocampal cellular composition and circuit function via epigenetic and transcriptional mechanisms. In adult F1_{GCS} offspring, we observed an elevated proportion of immature granule neurons, enhanced glutamate signaling, and reduced correlations between hippocampal gene expression and emotional-disorder-associated pathways. Together, these cellular and molecular shifts provide a mechanistic basis for the consistent alleviation of anxiety- and depression-like behaviors.

Moreover, as presented in the revised manuscript (Fig. 5B; Page 19, Lines 17–19), snRNA-seq data corroborated by qRT-PCR confirmed significant upregulation of choline-pathway genes—including *Chrna3*, *Creb3l2*, *Itp3*, and *Pla2g4b*—in the hippocampus of F1_{GCS} mice at P0. This result offers direct molecular evidence for the biological impact of gestational choline supplementation.

4. Why was only the right hippocampus dissected for analysis? What were the coordinates and size of the sampled tissue? Could tissue size or the number of neurons and glia impact the analytical results? The tissue sampling method is critical for determining sampling bias, as some cells may not have been captured or analyzed, and different hippocampal subregions may exhibit significant regional variability.

Response: We apologize for the confused description in tissue sampling of the Materials and Methods section. For all bulk RNA-seq, we used the entire freshly dissected hippocampus for subsequent experiments. In the paired single-nucleus assays of snRNA-seq and snATAC-seq, the isolated hippocampal tissues were snap-frozen in liquid nitrogen for subsequent experiments. Specifically, the left hippocampus from each of six mice was processed for snRNA-seq library preparation, while the contralateral right hippocampus was used for snATAC-seq, enabling within-animal comparison across epigenomic and transcriptomic layers. During quality control, one sample was excluded from both datasets due to insufficient library yield or low nuclear integrity, resulting in five high-quality biological replicates per group. These methodological details have been added to the revised Materials and Methods section (Page 30, Lines 16–27).

5. Considering the low sample size at each developmental stage, what do the authors consider the biological variable for the analysis...the litter or the individual animal? Clarification on this point would be valuable.

Response: We thank the reviewer for the comments. In our analyses, each offspring mouse was treated as an independent biological replicate for the measured outcomes. However, we agree that the litter effect could also introduce variability. To prevent potential litter effects, we strictly limited testing to 1–2 male offspring per dam in the molecular and behavioral analysis. This point has been clarified in the “Mice and Gestational Choline Supplementation (GCS) Program” subsection of the Materials and Methods in the revised manuscript (Page 28, Lines 14–21).

6. Although the cell communication analysis is interesting, it is limited to the hippocampus and does not provide functional insight into behavior, which is influenced by other brain regions.

Response: We appreciate the reviewer’s insightful suggestion. The hippocampus, with its established neural circuitry, plays a critical role in emotional processing and stress responses (PMID: 28521127, 29397273, 28176756). However, other brain regions, including the medial prefrontal cortex (mPFC) and amygdala, also contribute significantly to the regulation of emotional behaviors. To address this limitation, we have expanded the Discussion section in the revised manuscript (Page

26, Lines 17–27) to explicitly note that the behavioral effects associated with gestational choline supplementation (GCS) may involve broader neural circuits beyond the hippocampus.

7. *The correlation of gene expression changes in the used animal model with databases related to Alzheimer's disease, bipolar disorder, insomnia, major depressive disorder, and schizophrenia may be of potential interest, but in this study it appears only partially justified and insufficiently explained. What was the rationale for including these comparisons? Were they driven simply by data availability, or was there a specific hypothesis related to nutritional intervention?*

Response: We thank the reviewer for the insightful comment regarding the rationale for comparing gene expression changes in our animal model with disease-associated signatures in the scDRS database. These comparisons were guided by both methodological rigor and biological plausibility, which may evaluate the effects of GCS on broader neurodevelopmental and psychiatric diseases besides emotional behaviors. Particularly, granule neurons in the hippocampus has been linked to various neurological and psychiatric disorders, including ADHD (PMID: 28219628), depression (PMID: 30528746), AD (PMID: 35411392), insomnia (PMID: 20598672), anxiety (PMID: 36680709), schizophrenia, and bipolar disorder (PMID: 40506851). Moreover, we systematically evaluated all brain-related disorders available in the scDRS database based on the neurodevelopmental origin hypothesis of psychiatric diseases, which posits that early-life nutritional and environmental factors can influence disease risk through long-term effects on development and behavior (PMID: 28220606). The complete methodological rationale and related discussion have been added in the revised manuscript (Page 22, Line 17; Page 34, Lines 4–8).

Reviewer #3 (Remarks to the Author):

In this manuscript, Wang and colleagues examine the impact of gestational choline supplementation (GCS) on the hippocampal development using a mouse model. The authors examined the transcriptional alterations and chromatin accessibility modulated by GCS at the single-nucleus level using multi-omics approaches, such as single-nucleus RNA sequencing (snRNA-seq) and single-nucleus assay for transposase-accessible chromatin sequencing (snATAC-seq). The author found that the transcriptome, intercellular communication, and chromatin accessibility of immature granule neurons were altered in the hippocampus of male offspring with GCS. Using bulk transcriptome analysis of the hippocampus, the authors also demonstrated that GCS modulates neuronal development and biological events during critical developmental periods. Furthermore, single-cell disease relevance analysis (scDRS) revealed an association between the gene expression profiles of immature granule neurons in the hippocampus of FIGCS mice and various diseases, including major depressive disorder and anxiety. The reviewer commended the study's robust experimental design and rigorous bioinformatic analyses, including integrative approaches. This manuscript represents a significant contribution to the field of developmental neuroscience, setting high standards for experimentation and analysis. It also emphasizes key biological and analytical considerations for such studies while introducing gene expression variability at the single-cell level as a potential explanation for how nutritional supplementation affects central nervous system development. However, the manuscript remains incomplete in some areas that could be improved.

Major concerns:

1) *Experimental design:*

a) In this study, pregnant female mice were fed a choline-supplemented diet from gestational day 9 until parturition. However, I did not find a description of the reason for the feeding period. The authors need to explain why they chose this feeding period.

Response: We thank the reviewer for this valuable input. Maternal choline supplementation was administered from embryonic day 9 (E9) until delivery. These intervention period commenced after the completion of major neural tube closure events (PMID: 25032496, 36504449) and represented an important window for epigenetic reprogramming (PMID: 28220606) and a key period for nervous system development (PMID: 28891434). This explanation has been added to the revised “Mice and Gestational Choline Supplementation (GCS) Program” section (Page 28, Lines 7–13).

b) The authors collected the hippocampus from F1 mice at P0, P10, and P30 for bulk RNA-seq, and at P60 for snRNA-seq and snATAC-seq. As mentioned above, there is no explanation for why the samples were collected at these time points. The authors should explain why they chose these time points to collect the hippocampus.

Response: We appreciate the reviewer’s insightful comment. The sampling time points were strategically selected to encompass key postnatal developmental stages: P0 (newborn), P10 (peak of synaptogenesis), and P30 (near maturity) for bulk RNA-seq to capture transcriptional dynamics throughout development. Since behavioral assessment requires a fully mature nervous system (PMID: 30201220), we conducted snRNA-Seq and snATAC-Seq at P60 to identify cell-type-specific molecular mechanisms that underlie the stable behavioral differences observed in adulthood (Page 31, Lines 14–16). The rationale for the selection of these time points has been provided in the “Materials and Methods” section, under the “RNA-Sequence (RNA-Seq) and bioinformatics processing of the data” (Page 31, Lines 1–4).

2) Lacking the experimental evidence of cell proportion change:

The authors demonstrated an increased proportion of immature granule neurons and a decreased proportion of mature granule neurons in the hippocampus of F1 GCS mice at the gene expression level. Has a previous study investigated the proportion change modulated by GCS more direct method such as immunofluorescence staining of marker proteins? If not, the authors should consider performing the experiments showing the proportion change of granule neurons (mature to immature state) modulated by GCS at P60 to reinforce the conclusion.

Response: We thank the reviewer for this comment. We have performed validation using both Western blotting and quantitative real-time PCR (qRT-PCR) with established markers for immature and mature granule neurons in the revised manuscript. These results confirm the increase in the proportion of immature granule neurons at both P0 and in adulthood following gestational choline supplementation (Fig. 3B–E; Page 10, Lines 20–25), and the results have been incorporated into the revised manuscript to strengthen our conclusions.

3) Consider the sex differences:

In this study, the authors examined the effects of GCS on the hippocampal development in male F1 mice. Why were only male offspring analyzed? Many models nowadays exhibit dysmorphic responses, so it would greatly enrich the work if female offspring were also studied. If not, the authors should explain their reasoning in the manuscript.

Response: We sincerely appreciate the reviewer's valuable comment. This approach was adopted to reduce gender heterogeneity and minimize potential confounding effects associated with hormonal fluctuations during the estrous cycle in females (PMID: 39002829). We fully acknowledge that analyzing only one sex is a limitation, and we have discussed this point in the revised manuscript (Page 27, Lines 20–25).

Minor concerns:

1) In the title of the manuscript, I would like you to consider the inclusion of “hippocampus (or hippocampal)”.

Response: We appreciate the reviewer's suggestion. The title has been revised to: Gestational choline supplementation regulates hippocampal granule neuron development and emotion-like behavior (Page 1, Lines 1–2).

2) *It is not clear if the hippocampus was dissected after or before freezing the brain. Detailed method for dissection of the hippocampus should be clarified to ensure reproducibility.*

Response: We thank the reviewer for pointing out this issue. Details regarding the hippocampal tissue sampling have been provided in the “Materials and Methods” section under “Hippocampal dissection” (Page 30, Lines 15–27). Briefly, the dissection for bulk RNA-Seq, snRNA-Seq, and snATAC-Seq was performed independently at comparable circadian times. Mice were deeply anesthetized with isoflurane (pedal reflex absent confirmed) prior to decapitation. The whole brain was extracted and transferred to ice-cold PBS, followed by removal of the brainstem and cerebellum via a coronal cut at the midbrain under a stereomicroscope. For all bulk RNA-seq, we used the entire freshly dissected hippocampus for subsequent experiments. In the paired single-nucleus assays of snRNA-seq and snATAC-seq, the isolated hippocampal tissues were snap-frozen in liquid nitrogen for subsequent experiments. Specifically, the left hippocampus from each of six mice was processed for snRNA-seq library preparation, while the contralateral right hippocampus was used for snATAC-seq, enabling within-animal comparison across epigenomic and transcriptomic layers. During quality control, one sample was excluded from both datasets due to insufficient library yield or low nuclear integrity, resulting in five high-quality biological replicates per group.

3) *In Fig. S3A, asterisk is disappeared.*

Response: We thank the reviewer for highlighting this issue. In response, asterisks have been added to the relevant figures in the revised manuscript to enhance clarity and facilitate interpretation (Fig. S5A).

4) *Page 24, Line 5, “The time in open and closed arm were scored” should be revised as “The time spent in light and dark chamber were scored”.*

Response: We thank the reviewer for pointing this out. The text has been revised accordingly in our manuscript (Page 30, Line 13).

Response to the Reviewers:

Reviewer #1 (Remarks to the Author):

In this revised manuscript, Xiaohui Shi et al. have performed additional experiments and analyses to address the points raised by me and my co-reviewers during the first round of review. The authors have also undertaken substantial editing of the text. Overall, these changes have resulted in a significantly improved and clearer manuscript. Nevertheless, several aspects still require further attention. Below, I refer strictly to the points I raised in the first round of review and leave the assessment of my colleagues' comments to the other reviewers.

1. FDR correction across omics datasets

The consistent application of FDR correction across all omics datasets represents an improvement (even though the fold-change threshold remains relatively low). However, the authors need to ensure that this change is reflected consistently across all relevant figure panels and corresponding figure legends. For example, in Figure 2 (F–H), Figure 3 (F–G), and Figure S4 (A–C), the y-axis is still labeled as “ $-\log_{10}(p \text{ value})$ ” rather than “ $-\log_{10}(\text{FDR-corrected } p \text{ value})$ ” or “ $-\log_{10}(q \text{ value})$ ”.

Response: We sincerely thank the reviewer for this insightful comment. In the revised manuscript, the axis labels in Figures 2F–H, 3F–G, S5A–C and 6D have been uniformly updated to “ $-\log_{10}(q \text{ value})$ ” to consistently reflect the application of FDR correction across all omics datasets.

2. Statistical analysis of newly added experimental data

For the newly added experimental data presented in several figures (Figure 3B–E, Figure S2E, Figure S3, Figure 4F, Figure 4H, and Figure 5B), the authors should apply non-parametric statistical tests (e.g., Mann–Whitney tests) when $n = 3$ per group. The use of t-tests assumes normality, which cannot be reliably assessed with such small sample sizes. That said, I anticipate that the overall interpretation of the results will remain largely unchanged.

Response: We greatly appreciate the reviewer's valuable suggestion concerning the statistical analyses. We have reanalyzed the newly included experimental data, including Fig.3B–E (Page 13, Lines 4–9; Page 10, Lines 23–24), Fig. S3E (Page 16, Lines 4–5; Page 14, Lines 8–11), Fig. S4 (Page 16, Line 10), Fig. 4F, Fig. 4H (Page 18, Lines 14–23), and Fig. 5B (Page 22, Line 7), using the non-parametric Mann–Whitney U test and updated the corresponding figure legends and main text accordingly. As anticipated, the overall interpretation of the results remains robust: among the 28 genes validated at postnatal day 0 (P0), 26 exhibited significant differences ($p < 0.05$); and five of the eight genes validated at P60 reached significance.

3. Interpretation of persistent differentially expressed genes

*With regard to the two genes—*Lpar1* and *Ttr*—that show differential expression emerging at P10 and persisting through P60, it would be helpful to clarify whether these genes fall into the category of what the authors term “differentially accessible genes (DAGs).” If so, altered chromatin accessibility could represent a plausible mechanism of epigenetic memory linking early exposure to later outcomes. It would therefore be valuable to reference these two genes explicitly in the Discussion, particularly if they have known biological relevance to the observed phenotypes.*

Response: We thank the reviewer for this insightful suggestion. Integrative analysis of our multi-omics data identified *Lpar1* as a DAG with increased chromatin accessibility in mature granule neurons and astrocytes at P60. In contrast, *Ttr* showed no significant change in chromatin accessibility at P60, suggesting regulation by upstream transcriptional or post-transcriptional factors rather than direct epigenetic remodeling. We have incorporated these points into the revised Results section (Page 19, Lines 8–10), correspond figure and figure legends (Fig. S5H, Page 21, Lines 3–5), and Discussion section (Page 27, Lines 7–12) as follows: “Notably, bulk transcriptomic analysis indicated that *Lpar1*, a DAG in mature granule neurons and astrocytes, exhibited differential expression beginning at P10 and persisting through P60 in the hippocampus of F1_{GCS} mice, suggesting that GCS induced enduring epigenetic remodeling at this locus. Given its established roles in neuronal migration, myelination, and stress responsiveness (PMID: 34385905), sustained upregulation of *Lpar1* may reflect a mechanism of transcriptional memory that contributes to the observed anxiolytic and antidepressant phenotypes in F1_{GCS} mice”.